# Daily rainfall variability controls humid heatwaves in the global tropics and subtropics

Lawrence S. Jackson [1] ✉, Cathryn E. Birch [1], Guillaume Chagnaud[2], John H. Marsham [1] & Christopher M. Taylor [2,3]

Humid heatwaves are a growing risk to human and animal health, especially in tropical regions. While there is established research on dry-bulb temperature heatwaves, greater understanding of the meteorological drivers of extreme humid heat is urgently needed. In this study, we find that recent rainfall is a key control on the occurrence of humid heatwaves in the tropics and subtropics and its effect is regulated by the energy- or moisture-limited state of the land surface. In moisture-limited environments, heatwaves are likely during, or immediately after, enhanced rainfall. In energy-limited environments, heatwaves are likely after suppression of rainfall for two days or longer. The nature of the threat to health from heat stress varies by environment. It depends on local adaptation to temperature or humidity extremes, as well as vulnerability to absolute or anomalous extremes. Early warning systems, which reduce exposure and vulnerability to weather extremes, can benefit from this understanding of humid heatwave drivers, highlighting the possibility of predicting events using satellite-derived rainfall and surface moisture data.

Periods of combined high temperature and humidity result in heat stress, which can have a significant impact on human and animal health[1] and restrict economic activity, such as outdoor work[2]. Studies on human physiological responses to humid heat extremes suggest a potential limit to human adaptability to such conditions. After a few hours of exposure to wet-bulb temperature (Twb) of 35 °C, effective thermoregulation may become impossible[3]. Recent empirical studies suggest that health impacts occur at around 31 °C for young, healthy individuals[4], and likely at much lower values for vulnerable groups such as the very young and the elderly[5]. The focus of this study is on the physical processes involved in the occurrence of humid heatwaves, with particular attention to the role of rainfall variability.

Humid heat extremes have intensified since 1979[6,7] and become more frequent[6,8]. Some regions in the subtropics, such as coastal areas around the Persian Gulf, have already experienced short-lived Twb extremes above 35 °C[6,9]. Climate change is driving the occurrence of more extreme humid heat events, as seen in South and Southeast Asia

in April 2023[10]. Most importantly, humid heat stress is projected to become more severe under future climate change[1,11–17].

Along with an increased awareness of the risks posed by humid heatwaves there has been a growing focus on their underlying processes. In regions such as the southern Persian Gulf, north-central Pakistan, eastern South Asia, and the western Amazon, strong surface evaporation and the inhibition of convection by atmospheric stability play a key role in the occurrence of humid heat extremes[18]. A study using a pan-Africa convection-permitting climate model[19] found that humid heat is often associated with rainfall, with approximately one-third of humid heatwaves starting on wet days. In addition to rainfall, other factors exerting a control over humid heatwaves in all but the most humid regions of Africa include increases in cloud cover, surface longwave radiation, and surface evaporation. However, in highly humid regions such as the Congo Basin, factors regulating dry-bulb temperature are more significant than rainfall[19].

[1]School of Earth and Environment, University of Leeds, Leeds, UK. [2]UK Centre for Ecology and Hydrology, Wallingford, UK. [3]National Centre for Earth Observation, Wallingford, UK. ✉e-mail: l.s.jackson@leeds.ac.uk

Unlike dry heatwaves, extreme Twb events often occur during the monsoon season rather than prior to it, and the association between rainfall and humid heat varies with the local near-surface moisture climatology[20]. In parts of South Asia with relatively low specific humidity, Twb extremes are more likely during periods of enhanced rainfall, whereas Twb extremes are more likely during suppressed rainfall in regions of higher specific humidity[20]. In the arid and semi-arid regions of the global tropics and subtropics, light rain sustains surface evaporation and is associated with more frequent and more intense wet-bulb globe temperature extremes during the hottest four months of the year[21]. Irrigation has been shown to be important for humid heat extremes in India[22] and the northern plains of China[23], but there is no consensus on the nature of its control[24,25], partly due to different regional impacts of irrigation on moisture transport[26]. Coastal regions may be more at risk from extreme humid heat events[18] and, in India for instance, from compound humid heat and rainfall events[27]. Land-atmosphere feedbacks intensify and help the propagation of dry heatwaves[28], and increasing ecosystem water limitations are linked with intensified future dry heat extremes[29]. A widespread positive correlation between Twb and soil moisture[30] suggests that land-atmosphere feedbacks also play a key role in humid heat extremes. Much of the existing research on humid heat processes, however, is limited to regional-scale mechanisms, and the understanding of underlying processes at the scale of the global tropics and sub-tropics remains poorly understood[1,17].

We have two aims for this study. Our first aim is to characterise the relationship between occurrences of humid heatwaves and rainfall over land across the global tropics and subtropics. Our second aim is to test the hypothesis promulgated by Ivanovich et al.[20] and Zhang et al.[21] that rainfall regulates humid heat extremes through surface moisture flux processes that vary between energy- and moisture-limited environments[31,32]. Importantly, our study focuses on the global tropics and subtropics, whereas Ivanovich et al.[20] focus on South Asia. Additionally, our study addresses both the promotion and suppression of humid heat extremes, whereas Zhang et al.[21] focus more on the enhancement of extremes.

We use the daily mean Twb as our primary measure of humid heat because it is a useful metric for understanding humid heat dynamics in climate science. It combines both temperature and humidity and has similar thermodynamic properties to equivalent potential temperature ($\theta e$), making it robust for analysis on regional scales[33]. Furthermore, Twb is relevant to human heat stress as it provides an upper temperature limit for healthy, well-acclimatised human beings who have access to drinking water, shade, and a strong breeze[1]. Nevertheless, we acknowledge its limitations in representing human heat stress, particularly in relation to health outcomes. The 35 °C threshold for Twb[3] may substantially underestimate the range and severity of heat-related outcomes because it assumes specific conditions that are unlikely to apply in all contexts[34]. Additionally, Twb is a compound metric, and distinguishing between the contributions of temperature and humidity is crucial for understanding their distinct impacts, particularly because high Twb values are often driven by anomalously high humidity, with temperature anomalies playing a secondary role[34]. Furthermore, epidemiological studies have found little evidence for the impact of humidity on heat–health outcomes in some contexts[35] although epidemiological evidence is likely to develop under increasingly extreme future temperatures driven by climate change.

In this study, we prepare a dataset of humid heatwaves using ERA5 reanalysis data, validated against surface station observations, and identify distinct heatwave events using a time-space connected components method that has previously been used in the identification of heatwaves[36,37]. We use the local 95th percentile of daily mean Twb to capture humid heat events that are high for the locality, with an absolute minimum of 24 °C to exclude events likely low risk for most people[4,38] while ensuring sufficient data for statistically significant

results. The 95th percentile of daily mean Twb is calculated locally for each grid cell, accounting for regional variations, and is based on all days of the year, inherently capturing the seasonal cycle. We prepare a dataset of enhanced and suppressed rainfall days using GPM-IMERG data and assess the significance of the relationship between humid heatwaves and rainfall in years 2001–2022. At the time of each heatwave, we classify the surface environments as energy- or moisture-limited using the Ecosystem Limitation Index[39], an index of surface energy- and moisture-limitation that has previously been used in the study of dry heat extremes[29]. We find that there is a statistically significant relationship between humid heatwaves and daily rainfall across tropical land areas. Exploring the extent to which this relationship is explained by energy- or moisture-limitations shows the key role of surface evaporation of recent rainfall in many regions of the tropics and subtropics.

## Results

### Humid heatwaves and seasonal rainfall

The occurrence of humid heatwaves is widespread across the global tropics and subtropics (Fig. 1a). These heatwaves occur in monsoon regions such as West Africa, India, East China, and north Australia, and in equatorial regions such as the Amazon, the Congo basin, and the Maritime Continent. Most of these regions experience 1–2 heat waves per year. In general, the frequency of occurrence decreases sharply at the northern and southern extent of monsoon regions due to climatologically low humidity (e.g., West Africa, Australia) and in regions at high elevation due to lower dry-bulb temperatures at altitude (e.g., areas within the Himalayan region in India and inland East Africa). The heatwaves are relatively small in size. The minimum size of the heatwaves is approximately 1900 km² (by definition), while the median size is approximately 4400 km².

The median duration of humid heatwaves is typically between 3 and 6 days (Fig. 1b). Regions with the longest heatwave durations include the equatorial east coast of Africa, the Indian subcontinent, and eastern China. In contrast, heatwaves are relatively short in South America, West and Central Africa, and Southeast Asia. Heatwave intensity, represented by the 95th percentile of daily mean Twb on heatwave days, typically ranges between 24 °C and 27 °C (Fig. 1c). Regions with the most intense heatwaves include the Red Sea and Persian Gulf coastlines, Pakistan, northern India and Bangladesh, and northeast China, where the daily mean Twb 95th percentile ranges between 28 °C and 31 °C. Daily maximum Twb can be much higher[18,40].

Humid heatwaves and regional rainfall exhibit a strong seasonal cycle (Fig. 1d, e). Over the western and central regions of the Amazon basin, heatwaves most frequently occur during October-November, marking the transition from the dry to wet season. In equatorial regions (10°S–10°N), humid heatwaves occur at the time of the seasonal migration of the tropical rain-belt, particularly in March-May over eastern South America, Africa, and western Maritime Continent. Regions such as India, Southeast Asia, East and West Africa, and North Australia are farther from the equator and experience strong monsoon seasons (e.g., peak rainfall occurs in July-August over much of India and the West African Sahel). In these regions, humid heatwave occurrence appears to synchronise with monsoon seasons.

To show the relationship between humid heatwaves and seasonal rainfall at the monthly timescale, Fig. 1f shows the number of months separating the seasons of peak rainfall and peak humid heat (i.e., the difference between Fig. 1e and d). In regions beyond latitudes 10°S–10°N that have strong seasonal monsoons, the seasons of rainfall and humid heat are closely synchronised. These are regions shaded cream in Fig. 1f (e.g., West African Sahel). However, it is important to note that this qualitative association between seasonal rainfall and humid heat does not establish a causal relationship between the two phenomena. Furthermore, there are many regions exposed to humid heat in which peak seasonal rainfall and humid heat do not coincide.

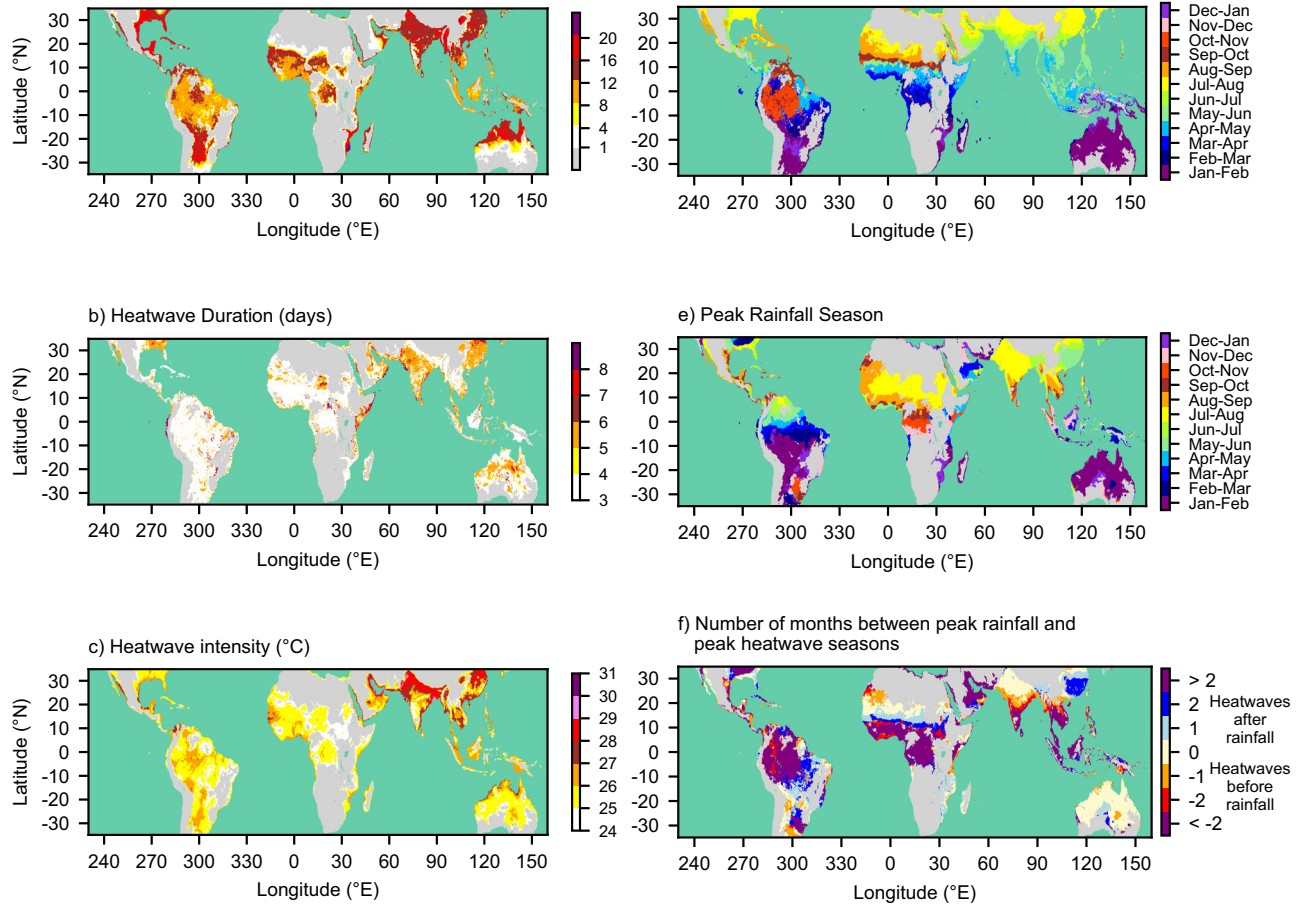

**Fig. 1 | Humid heatwaves over land in the global tropics and subtropics, and the seasonality of rainfall and humid heatwaves. a** The frequency of heatwaves in each grid cell expressed as the number per decade. **b** The median duration of heatwaves in each grid cell. **c** Heatwave intensity in each grid cell, represented by the 95th percentile (p95) of daily mean wet-bulb temperature (Twb) on heatwave

days. **d** The two-month period during a calendar year in which most heatwaves occur. **e** The two-month period during a calendar year in which the highest rainfall occurs in GPM-IMERG (2001 to 2022). **f** The number of months between the month of peak rainfall and the month of peak heatwave occurrence. Regions with no heatwaves are shaded grey.

These regions, where heatwaves can occur before or after seasonal rainfall, include many monsoon areas between 10°S and 10°N and are shaded purple in Fig. 1f.

### Rainfall variability as a driver of humid heatwave occurrence

In the previous section, we established a link between the timing of humid heatwaves and of rainfall at the seasonal timescale. As a step towards a process-based understanding, we now examine the relationship on the daily timescale in each grid cell using months in which at least one humid heatwave occurred. We categorised each day into one of four groups based on whether rainfall was higher or lower than a minimum threshold of 5 mm/day in a two-day window, and whether a heatwave started or not. The groups with higher rainfall (High_Rain) include days when rainfall exceeds 5 mm/day on that day, or the previous day, or on both days. The groups with lower rainfall (Low_Rain) include days when rainfall does not exceed 5 mm/day on both that day and the previous day. We used counts of the number of days in each group to calculate empirical probabilities for heatwaves starting on High_Rain days or Low_Rain days (see Methods for more details).

In Fig. 2a, we map the relative risk (the ratio of the two probabilities) in those grid cells where there is a statistically significant difference between the probabilities. Blue shades represent grid cells where heatwaves are more likely on High_Rain days, and red shades represent grid cells where heatwaves are more likely on Low_Rain days. There is a clear geographic pattern which resembles the Köppen-

Geiger classification of climate zones[41]; humid heatwaves are more likely to occur on High_Rain days in arid regions (Köppen-Geiger Zone B). In contrast, humid heatwaves are more likely to occur on Low_Rain days in tropical and equatorial regions (Köppen-Geiger Zone A).

To examine the role of surface energy- and moisture-limitations in modulating the relationship between rainfall and humid heat, we calculated the Ecosystem Limitation Index[39] (ELI) daily for each grid cell. The ELI is calculated as the difference between: (i) the correlation of surface latent heat flux with near-surface soil moisture; and (ii) the correlation of surface latent heat flux with downwelling shortwave (SW) radiation at the surface using variables from ERA5 reanalysis and over a 7-day rolling window (see Methods). Positive values for the ELI mean that soil moisture exerts a stronger control on surface evaporation than downwelling surface SW radiation, signifying that moisture limitation is greater than the energy limitation. Hereafter, we refer to this as moisture limitation. In contrast, negative values indicate greater energy limitation than moisture limitation, which hereafter we refer to simply as energy limitation. In Fig. 2b, the ELI shown is a heatwave-composite mean of values taken two days before the start of humid heatwaves. This ELI, therefore, represents surface energy- and moisture-limitation immediately before and at the start of humid heatwaves and is not necessarily indicative of the local seasonal climatology.

The scatter plot of grid cell data in Fig. 2c shows that humid heatwaves fall into one of four rainfall-ELI regimes (see Methods).

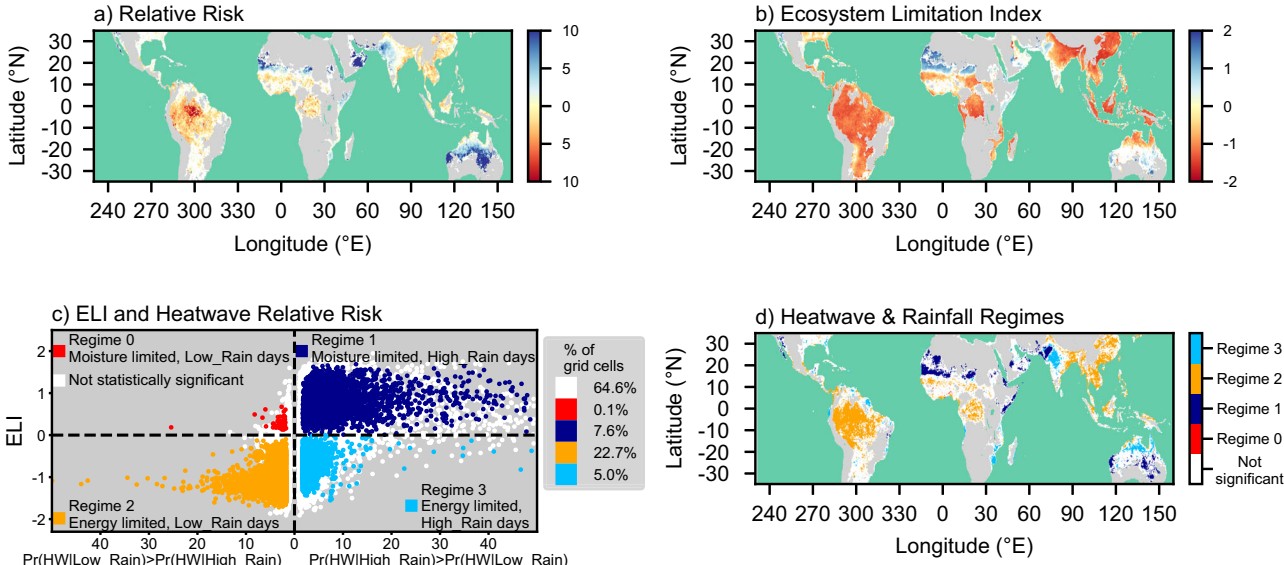

**Fig. 2 | Linking the occurrence of heatwaves to rainfall (2001-2022). a** Relative risk of heatwave occurrence on High_Rain (blue) and Low_Rain (red) days. **b** The Ecosystem Limitation Index (ELI): moisture-limited regions (ELI > 0), energy-limited regions (ELI < 0). **c** Scatter plot of ELI against relative risk and the percentage of grid cells in each regime, among all land grid cells between latitudes 35°N–35°S where humid heatwaves occurred in 1993–2022 (inset). **d** Map of the heatwave and rainfall regimes defined in (**c**). In (**a**, **b**, **d**) no humid heatwaves were identified in regions shaded grey. In all panels, regions not significant at the 5% level are shaded white.

Regime 0, in which heatwaves occur in moisture-limited environments and are more likely on Low_Rain days, is relatively rare (0.1% grid cells, red shading). Its grid cells have relatively small ELI values (median 0.27) which could potentially result from noise in other regimes. Therefore, Regime 0 is not examined further, along with grid cells that are not statistically significant (66% of the grid cells, white shading).

Figures 2a, b show that heatwaves occurring in moisture-limited environments generally occur on High_Rain days (Regime 1). These heatwaves occur in 8% of land grid cells between latitudes 35°N and 35°S where humid heatwaves occurred during 1993–2022 (Fig. 2c, d, dark blue shading). Heatwaves occurring in energy-limited environments generally occur on Low_Rain days (Regime 2) and account for 23% of the grid cells (Fig. 2c, d, orange shading). For grid cells with moderate relative risk values (i.e., relative risk <10), there is a strong linear relationship between relative risk and ELI (Pearson correlation coefficient +0.74). Grid cells on the right-hand side of Fig. 2c with relative risk values > 10 (i.e., where Eq. (3) applies; see "Methods") have a mean ELI of +0.92 ± 0.01 (±1 standard error) and are clearly moisture-limited. Similarly, grid cells on the left-hand side of Fig. 2c with relative risk values > 10 (i.e., where Eq. (4) applies; see Methods) have a mean ELI of −1.14 ± 0.02 and are clearly energy-limited.

Grid cells with a negative ELI and a positive relative risk are assigned to Regime 3 (Fig. 2c). This regime occurs in parts of western India, Mozambique, and northern Australia, where the local environment is energy-limited according to the ELI, yet heatwaves are more likely on High_Rain days (Fig. 2c, d, light blue shading). This apparent contradiction is explained in the next section, where we show that surface evaporation in Regime 3 is sensitive to both energy- and moisture-limitations.

## Contributions of temperature and specific humidity to humid heatwaves

Figure 3 shows composite time series centred on the first day of each heatwave for Twb, 2 m temperature, and 2 m specific humidity. Grid cell heatwave composites are constructed by averaging daily mean time series spanning 7 days before to 7 days after the start of each heatwave. For grid cells where the heatwave-rainfall relationship is statistically significant (Fig. 2), the composites are grouped into the regimes shown in Fig. 2d. The composite for each regime is then calculated as the average of grid cell composites within that regime. The heatwaves in our analysis last a minimum of 3 days (days 0–2) and have a median duration of 4 days (days 0–3). By day 7, ~79% of the heatwaves have dissipated. To maintain consistency in the number of data points across all days, the composites for days 3–7 include data from both ongoing and dissipated heatwaves.

Regions in Regime 1 are climatologically hot and dry, with relatively modest levels of Twb prior to the heatwaves, consistent with the moisture-limited nature of Regime 1 (Fig. 2c). Anomalies in temperature and specific humidity grow strongly starting two days before the heatwaves, coinciding with days of High_Rain immediately before and at the start of the heatwaves. In Regime 1, humid heatwaves result from large positive specific humidity anomalies, partly offset by negative 2 m temperature anomalies (~−2 °C).

Regime 2 is energy-limited (Fig. 2c) and is climatologically cooler and more humid than Regime 1. Anomalies in temperature and specific humidity grow during the first two days of the heatwaves following two days of Low_Rain. Unlike Regime 1, humid heatwaves in Regime 2 result from positive anomalies in both 2 m temperature and specific humidity. Although heatwaves have the largest absolute Twb in Regime 2, the temperature and specific humidity anomalies are smaller than those in Regime 1.

Regime 3 is climatologically similar to Regime 2, although slightly hotter and less humid. In contrast to Regime 2, heatwaves in Regime 3 primarily result from positive specific humidity anomalies. There is a statistically significant increase in 2 m temperatures from day −1 to day 1, although the anomalies during the heatwave are relatively small (<0.5 °C).

## The controlling role of surface energy- and moisture limitations

To elucidate the mechanisms underpinning each heatwave regime, Fig. 4 presents composite time series centred on the first day of each heatwave for several surface and atmospheric variables.

Rainfall anomalies on day −1 and day 0 differ between the regimes by design. Differences in anomalies on days −7 to −2 and days 1 to 7, however, provide insight into the role of rainfall variability in the occurrence of humid heatwaves. In Regime 1 (moisture-limited), daily

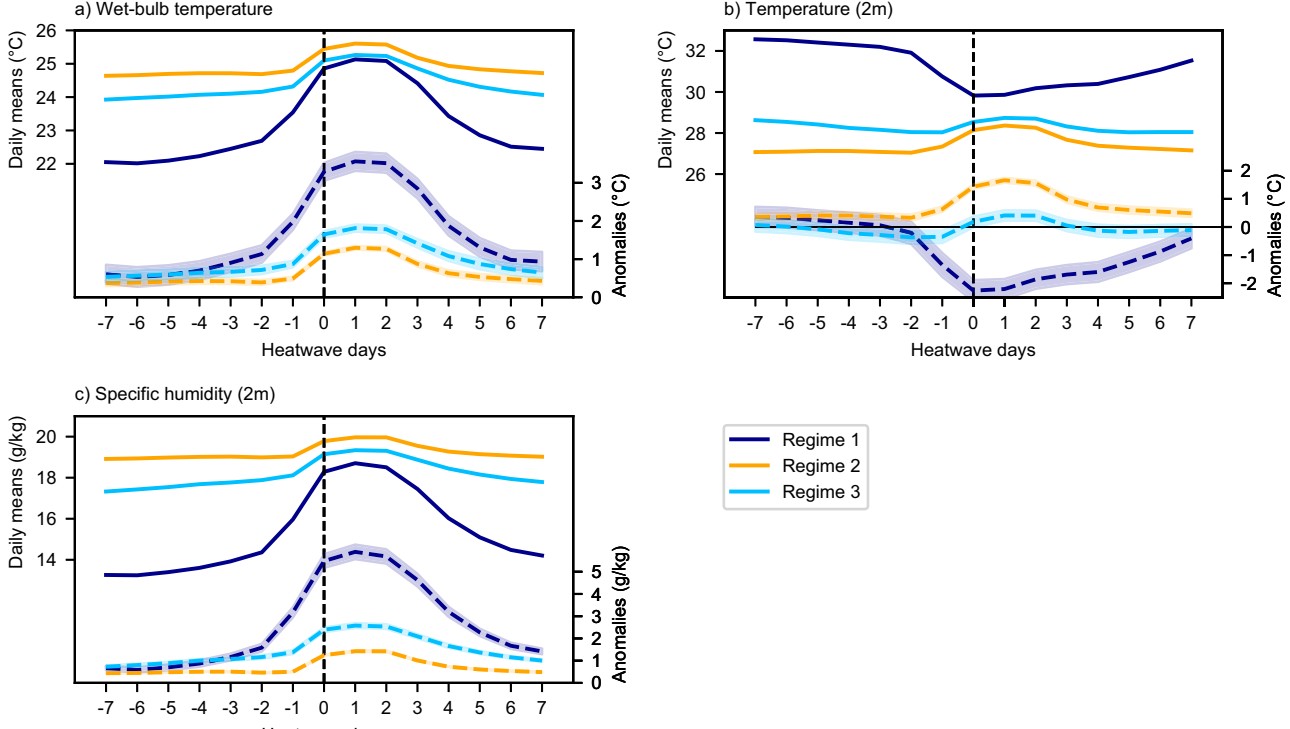

**Fig. 3 | Heatwave composite time series from 7 days before the start of each heatwave (day -7) to 7 days after the start of each heatwave (day 7). a** Wet-bulb temperature (Twb), (**b**) temperature (2 m), and (**c**) specific humidity (2 m). The time series runs from 7 days before the start of each heatwave (day −7) to 7 days after the start of each heatwave (day 7). Day 0 is the heatwave start day. Solid lines and the y-axis on the left of each panel show composite daily mean values. Dashed lines and the *y*-axis on the right show composite anomalies from the local 1993–2022 daily mean climatology. The shading about the dashed lines shows 95% confidence intervals for the daily anomalies.

mean rainfall is relatively low. The rainfall anomaly increases strongly in the two days before the heatwave, and is enhanced just prior to the start of the heatwave (day −1). For Regimes 2 and 3 (energy-limited), daily mean rainfall is greater three or more days before the start of a heatwave. In Regime 2, rainfall decreases sharply two days before a heatwave, it is suppressed at the start of a heatwave, and returns to its daily climatology over the next three days. In Regime 3, positive rainfall anomalies occur in the run up to a heatwave (days −6 to −1). Rainfall decreases to its climatology at the start of a heatwave followed by a peak in rainfall three days later when the heatwaves start to dissipate (on day 3).

In summary, rainfall variability at the start of a humid heatwave can be characterised as enhanced (Regime 1), suppressed (Regime 2), or neutral (Regime 3).

### Regime 1 (ELI moisture-limited and enhanced rainfall)
Prior to rainfall, the environment in these regions is relatively dry characterised by low rainfall, low total column water vapour (TCWV), low soil moisture, and a relatively deep boundary layer (Fig. 4). Rainfall immediately before and/or during the first days of a heatwave, increases soil moisture and promotes humidification of the boundary layer through a strong increase in surface latent heat flux. Specific humidity may also be enhanced aloft by moisture convergence associated with the rain-bearing storms. The overall increase in surface latent heat flux is accompanied by a decrease in surface sensible heat flux, which is characteristic of moisture-limited environments. This decrease in surface sensible heat flux reduces the depth of the boundary layer, concentrating specific humidity near the surface. The increase in atmospheric specific humidity enhances downwelling longwave radiation at the surface, although this increase in surface radiation is more than offset by attenuation of downwelling SW

radiation due to increased cloud cover associated with the storms. The associated changes in net surface radiation fluxes are shown in Fig. S1.

### Regime 2 (ELI energy-limited and suppressed rainfall)
The environment in these regions is wetter, and the supply of energy and moisture from the surface to the near-surface air differs from regions that are moisture-limited. Prior to a heatwave, rainfall and TCWV are greater than in the moisture-limited regime, and wetter soils support greater latent heat fluxes at the surface. Cloud cover associated with the rainfall prior to heatwaves attenuates downwelling SW radiation at the surface.

The humid heatwaves in these regions are associated with suppressed rainfall from the day before the start of a heatwave to 2 days after the start of the heatwave. The near-surface soil remains moist and the atmosphere humid although changes from climatology are very small. Positive anomalies in downwelling SW radiation, which exceed $25\,Wm^{-2}$, promote an increase in surface latent heat flux. Changes in surface sensible heat flux and boundary layer height are very small. Taken together, these are changes characteristic of energy-limited environments, due to climatologically wetter soils.

### Regime 3 (ELI energy-limited and neutral rainfall)
Prior to the onset of humid heatwaves (day −7), the environment in these regions resembles that of Regime 2 in terms of rainfall, TCWV, and downwelling SW radiation at the surface. In Regime 3, however, the surface is drier with less near-surface soil moisture.

The critical difference between Regimes 2 and 3 is the response of surface latent heat fluxes to changes in surface moisture and energy, as indicated by the correlations between these variables (Table 1). Over the 15 days centred on the start of the humid heatwaves, Regime 3 has positive correlations for surface latent heat flux anomalies with both

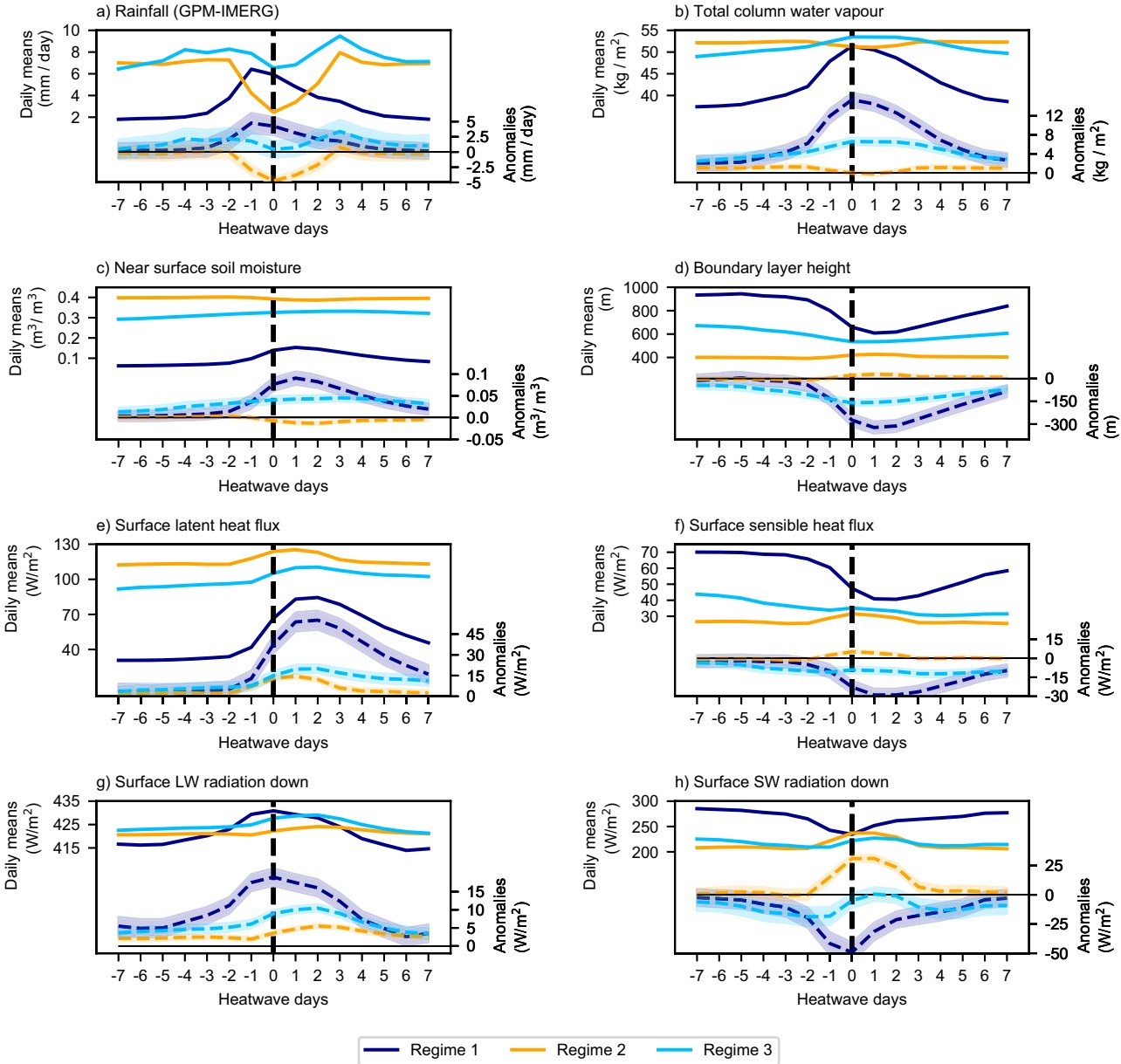

**Fig. 4 | Heatwave composite time series from 7 days before the start of each heatwave (day -7) to 7 days after the start of each heatwave (day 7). a** Rainfall, (**b**) total column water vapour, (**c**) near surface soil moisture, (**d**) boundary layer height, (**e**) surface latent heat flux, (**f**) surface sensible heat flux, (**g**) surface longwave (LW) radiation down, and (**h**) surface shortwave (SW) radiation down. Day 0 is the heatwave start day. Solid lines and the *y*-axis on the left of each panel show daily mean values. Dashed lines and the *y*-axis on the right show anomalies from the local 1993–2022 daily mean climatology. The shading about the dashed lines shows the 95% confidence intervals for the daily anomalies.

soil moisture anomalies (+0.88) and downwelling SW radiation anomalies (+0.54). In contrast, Regime 2 has a negative correlation between surface latent heat anomalies and soil moisture anomalies (−0.76) and a positive correlation between surface latent heat anomalies and downwelling SW radiation anomalies (+0.99). For comparison, Regime 1 (moisture-limited) has correlations of +0.96 between surface latent heat flux anomalies and soil moisture anomalies and −0.47 between surface latent heat flux anomalies and downwelling SW radiation anomalies.

In Regime 3, prior to the onset of a humid heatwave, surface evaporation and Twb are constrained by surface energy and moisture availability. Rainfall prior to a humid heatwave (days −7 to −2) supplies the necessary surface moisture, while a decrease in rainfall around the start of a heatwave (days −1 and 0) is accompanied by an increase in downwelling SW radiation, which increases surface energy. Together,

these changes overcome the initial constraints on surface energy and moisture, leading to an increase in surface latent heat flux. This, in turn, raises near-surface specific humidity and Twb. These changes are reinforced by a shallower boundary layer resulting from weaker surface sensible heat fluxes.

In summary, although Regime 3 is energy-limited according to the ELI, both energy- and moisture-limitations actually influence surface evaporation in this regime.

**The most extreme humid heatwaves**

To examine the most extreme humid heatwaves, we identify the heatwaves in each grid cell where the highest daily mean Twb during the event exceeds the 95th percentile of the highest daily mean Twb values calculated across all humid heatwaves in that grid cell (Fig. 5). Each grid cell is allocated to the same regime as used for all heatwave

events. Conditions during extreme heatwaves in each regime (solid circles) have a wide distribution of temperature and relative humidity, with substantial overlap between the three regimes (shading). Daily mean Twb during the most extreme events is between 27 °C and 28 °C in all three regimes. Heat Index[42] (HI) classifications of Extreme Caution (HI > ~33 °C) and Danger (HI > ~39.5 °C) are common in all regimes, although the very highest Twb and HI values occur in Regime 2. The mean maximum daily Twb conditions averaged over all humid heatwaves (crosses) are also similar across the regimes, ranging from 25 °C to 26 °C and HI 33 °C to 35 °C.

Despite the similar conditions experienced during extreme humid heatwaves across the three regimes, the climatological relative humidity and temperature, and their change in the build-up to events are very different. Locations within Regime 1 (moisture-limited) are climatologically hot and arid, with a climatological mean Twb during the heatwave season of 22 °C. A large increase in relative humidity is required for a humid heatwave to occur, and in the most extreme events, much of this occurs more than 7 days before the event. Locations within Regime 2 (energy-limited) are climatologically more humid, cooler in temperature, with a climatological mean Twb between 24 °C and 25 °C (during the heatwave season). A large increase in temperature is required for a humid heatwave to occur, which is accompanied by an increase in specific humidity to sustain the relatively high level of relative humidity, and in the most extreme events, much of this occurs more than 7 days before the event.

The elevated temperatures and relative humidity occurring seven days before the extreme humid heatwave events are likely driven by inter-annual variability. This variability, driven by large-scale environmental factors such as ENSO, preconditions the local environment in part through prior heatwaves. In 34% of extreme heatwave events, a preceding heatwave event occurred seven days before the onset of the extreme event.

Although the humid heat extremes are similar across the three regimes, the nature of the threat to human health from heat stress varies by regime. The threat depends on whether local populations are better adapted to temperature or humidity extremes, and whether they are more vulnerable to absolute or anomalous extremes. Additionally, the impact of a humid heat event at a given magnitude of Twb depends on its composition[34]. Extreme humid heat poses greater danger to human health when driven by high temperatures combined with moderate humidity, rather than high humidity with moderate temperatures[43].

## Discussion

Daily rainfall variability strongly controls humid heatwaves across substantial areas of the global tropics and subtropics in our analysis of ERA5 reanalysis and GPM-IMERG rainfall data. The response of humid heat to rainfall changes is modulated by the sensitivity of surface evaporation to soil moisture levels and downwelling SW radiation at the surface. In Regime 1, the surface environment is moisture-limited, and humid heatwaves are more likely immediately after enhanced rainfall. In Regime 2, the surface environment is energy-limited, and humid heatwaves are more likely during suppressed rainfall. In Regime 3, surface evaporation is sensitive to both downwelling SW radiation and soil moisture levels, making humid heatwaves more likely during near climatological levels of rainfall. Regime 0, characterised by moisture-limited conditions where heatwaves are more likely after low rainfall, is not explored in detail. It occurs in few grid cells, which have relatively low values for both relative risk and ELI.

We used ERA5 reanalysis data, which offers improved spatial and temporal resolution and provides better representation of tropical circulations and forecast skill compared to previous reanalysis products[44,45]. However, ERA5 has known limitations, particularly at regional scales[45,46]. To address this, we validated our analysis against hourly observational data from the HadISD dataset[47] (Fig. S2), pairing

**Table 1 | Correlations of anomalies in surface latent heat flux, soil moisture, and downwelling SW radiation for each regime**

| Regime | Correlation of surface latent heat flux anomalies with | |
|---|---|---|
|  | Soil moisture anomalies | Downwelling SW radiation |
| 1 | +0.96 | −0.47 |
| 2 | −0.76 | +0.99 |
| 3 | +0.88 | +0.54 |

each station with the nearest ERA5 land grid cell and reproducing the Twb heatwave composites that are shown for ERA5 data in Fig. 3a. The results demonstrate substantial agreement between Twb values from ERA5 reanalysis and observations, including consistent trends in the Twb composite time series.

A comparison of ERA5 reanalysis and the HadISD 2 m temperature dataset shows broad agreement, although ERA5 exhibits a cold bias in both Regimes 1 and 2 (Fig. S3). This bias may result from factors such as the averaging of ERA5 grid cell temperatures versus station point measurements, elevation differences, unresolved local topography[48], and the influence of land use/land cover on station observations. Additionally, ERA5 has been shown to underestimate daily maximum temperatures and extremes in some regions[49,50]. Unlike Regime 1, the cold bias in Regime 2 is consistent across all lead-lag times. Regime 2 primarily occurs in moist tropical environments, and in one such environment, tropical oceans, ERA5 has been found to underestimate temperatures and overestimate humidity, possibly due to deficiencies in relative humidity-dependent entrainment[51]. This mechanism could potentially contribute to the consistent bias in Regime 2.

In our assessment of statistical significance for the relationship between humid heatwaves and rainfall, the regimes are spatially coherent with clusters of significant grid cells clearly separated by areas of non-significant grid cells. However, the non-significant areas account for 66% of the land grid cells. Including these grid cells in our analysis of processes has little impact on the results. In Figs. 3 and 4, their inclusion makes the results for Regimes 2 and 3 more similar (Fig. S4 and S5). For Regime 1, the magnitude of the anomalies decreases slightly, likely due to a weaker signal-to-noise ratio in the non-significant grid cells. The lack of statistical significance may be due to several factors. First, the number of heatwave events may be insufficient for the underlying signal to emerge from noise introduced by sub-seasonal and inter-annual variability. Second, the identified heatwave-rainfall relationships may not prevail universally, with other processes potentially dominating humid heatwave dynamics (e.g., non-local surface evaporation and transportation of moisture[52]). Third, the relationship between heatwaves and rainfall may vary across different phases of the seasonal cycle. For example, in parts of peninsular India, the local environment may be moisture-limited during the pre-monsoon period but energy-limited after the onset of the monsoon. Indeed, in some regions, such as parts of India, central Africa, and South America, the humid heatwave season spans several months and could potentially accommodate seasonal variation in the heatwave-rainfall relation. While we conclude that our findings likely extend over parts of the white-shaded regions in Fig. 2d, a logical next step would be to extend the analysis to sub-seasonal timescales in the transition regions.

ELI captures the competing influences of soil moisture and surface SW radiation downwards on surface latent heat fluxes. To test our findings that surface energy- and moisture-limitations modulate the relationship between rainfall and humid heat, we defined a wet-bulb temperature energy-moisture limitation index (TwbLI). TwbLI captures the competing influences of soil moisture and surface SW radiation downwards on Twb. This index is analogous to the ELI but instead of using surface latent heat flux anomalies, Twb anomalies are used in the calculation of TwbLI (see "Methods"). The heatwave-rainfall

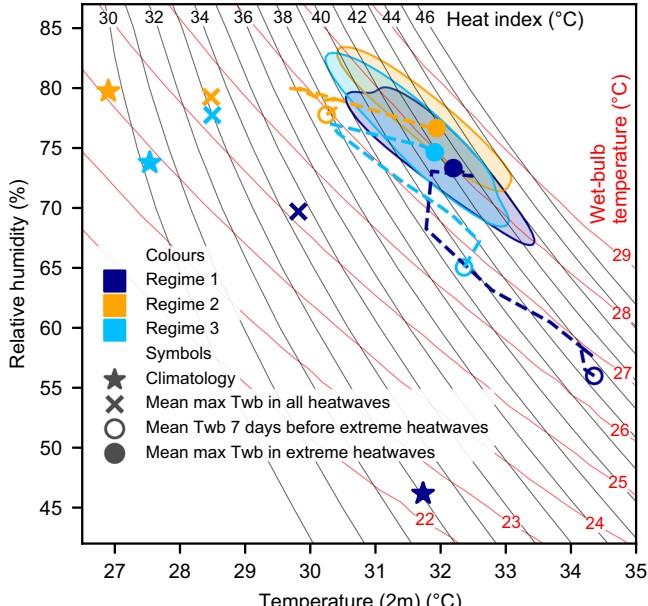

**Fig. 5 | Climatology (1993-2022) and humid heatwave conditions in each regime plotted on isopleths of wet-bulb temperature (Twb) and Heat Index.** The dashed lines show the daily time series from day −7 (open circle) to day +2 (solid circle) for the most extreme heatwaves (events with 10% highest Twb). The shaded areas show the distribution of temperature and relative humidity on the 2nd day of the extreme heatwaves. The shaded areas contain 50% of the extreme heatwave grid cells, and their boundary contour presents the 50th percentile. Extreme heatwaves are the heatwaves in each grid cell that have the highest 5% of Twb.

regimes identified using TwbLI (Fig. S6d) resemble those found using ELI (Fig. 2d), particularly for the moisture-limited Regime 1 and the energy-limited Regime 2. The consistency of heatwave-rainfall regimes between ELI and TwbLI suggests that energy and moisture limitations exert a dominant influence on Twb through their control of surface latent heat fluxes. Moreover, to first order, there is a strong correlation between the spatial distributions of ELI (Fig. 2b) and TwbLI (Fig. S6b) and the spatial distribution of Köppen-Geiger climate zones, particularly for zones A and B. With a projected future shift from energy- to moisture-limitation[39], rainfall variability could play an increasingly important role in future occurrences of humid heatwaves.

We tested the sensitivity of our results to changes in several key assumptions. Similar relative risk results were obtained with a 1 mm/day rainfall threshold, where 43% of land grid cells with heatwaves are significant (Figure S7a), compared to 45% for the 5 mm/day threshold (Fig. 2a). Defining rainy days as days of light rain (i.e. 1–10 mm/day) also produced similar results except the relative risk is weaker for heatwaves on light-rain days (Figure S7b). Excluding days of heavier rainfall weakened the relative risk results in all regions (Fig. S7c, d), suggesting heavy rainfall days (and potentially subsequent flooding) may be important for humid heat risk[53]. We also found similar results using ERA5 rainfall for years from 1993 to 2022 instead of GPM-IMERG (Fig. S8a). A threshold of 5 mm/day for enhanced rainfall days produced the clearest results for the relative risk of heatwaves occurring on High_Rain versus Low_Rain days. The 5 mm/day threshold is physically relevant: it is the median daily rainfall during 2001–2022 in GPM-IMERG in the humid heatwave seasons in regions with humid heatwaves as shown in Fig. 1. We interpret the threshold as representing above-average rainfall which is sufficiently heavy to promote surface evaporation.

We used a minimum threshold of 24 °C Twb, alongside the local 95th percentile of daily mean Twb, to identify hot, humid days and a sufficiently large number of heatwave events to support robust

statistical analysis. Repeating our relative risk analysis using the local 95th percentile of daily mean Twb and a higher minimum threshold of 26 °C Twb, a level above which vulnerable individuals can be significantly affected by heat stress[54], yielded a consistent spatial pattern of relative risk albeit with substantially reduced numbers of statistically significant grid cells due to the smaller numbers of heatwave events (Fig. S8b).

We used daily mean Twb for our analysis because it captures both daytime heating and night-time cooling, making it suitable for studying processes at daily timescales. It also accounts for accumulative humid heat over longer periods, including the significant contribution of night-time temperatures. However, since daily maximum Twb is often used in studies of humid heatwaves, we repeated our analysis with daily maximum Twb (using a higher minimum threshold of 24.75 °C). This yielded similar heatwave-rainfall regimes but with fewer statistically significant grid cells (Fig. S9).

While Twb is widely used and an appropriate measure for the study of humid heat dynamics in climate science, it has its limitations when applied to human heat stress. Twb does not fully incorporate the full range of human physiological responses to humid heat and provides a temperature threshold based on conditions ideal for maximal cooling through evaporation. Additionally, Twb does not account for incoming solar radiation and wind speed, factors relevant to human heat stress in external environments. Figure 4h shows significant changes in incoming SW radiation, largely driven by cloud cover variations, with these changes differing notably between the three regimes. Metrics such as the wet-bulb globe temperature[55] (WBGT) or the Universal Thermal Climate Index[56] (UTCI) are specifically designed to integrate these factors and are more suitable for assessing human heat stress responses. However, it is important to note that even these metrics have inherent limitations and may not fully capture all aspects of human physiological responses to extreme humid heat[57].

Although an assessment of human heat stress is beyond the scope of this study, we recalculated relative risk using the Rothfusz formulation of the Heat Index (Fig. S8c) and also 2 m dry-bulb temperature (Fig. S8d) to contextualise our results using two widely used temperature metrics. In energy-limited regions where heatwaves are more likely to follow suppressed rainfall, we obtain similar but more strongly significant results using the Heat Index or 2 m dry-bulb temperature. In contrast, in moisture-limited regions, the strength of the relationship between heatwaves and rainfall is much weaker. This supports our choice of Twb for this study and highlights differences in the sensitivity of different heat stress measures to humidity[20,25,58]. The Rothfusz formulation for the Heat Index is an empirical polynomial approximation based on Steadman's physiological data[59,60] and is not a comprehensive model of human heat tolerance. Its validity is limited under extreme temperature and humidity conditions, particularly beyond its original dataset[61,62]. Furthermore, both the Rothfusz Heat Index and 2 m dry-bulb temperature overlook factors such as incoming solar radiation and variations in wind speed, which are critical in outdoor environments[55].

Our results focus on the daily timescale, and an obvious next step would be to extend the analysis to hourly time scales. Extreme humid heat events can be short-lived, and processes at the diurnal timescale, such as land-sea breezes, low-level moisture-bearing monsoon winds, and moist convection, are likely of varying importance across locations in the three regimes identified in this study. A better understanding of humid heatwaves at the sub-daily timescale would inform near-real-time prediction. Future studies could focus on the more slowly evolving, longer timescale atmospheric processes, such as ENSO, that precondition the local environment for the subsequent development of humid heatwaves. Finally, projected changes in future rainfall patterns[63] and future shifts from energy- to moisture-limited

environments[29,39] highlight that the response of humid heatwaves to climate change is likely to be complex at the regional scale.

Our findings are a key step towards greater understanding of the meteorological processes underpinning humid heatwaves at the regional scale. They provide valuable insights for evaluating weather and climate models and assessing projected changes in humid heat under climate change. Crucially, these findings will inform the design of much-needed early warning systems for humid heat extremes. The key role of rainfall immediately preceding humid heatwaves highlights the potential for satellite observations of rain and soil moisture to enhance monitoring and early warning systems, especially in regions with limited surface weather station coverage.

## Methods
### ERA5 data
We use data from the European Centre for Medium-Range Weather Forecasts (ECMWF) fifth-generation reanalysis (ERA5)[64,65]. We use hourly data on the native ERA5 spatial grid (resolution of 0.25° × 0.25°). We exclude February 29th from leap years so that each year has 365 days in our study. All analysis is based on daily means calculated using hourly data based on 0-24 UTC days. We use data for years 1993 to 2022 for Fig. 1 and data for years 2001 to 2022 for Figs. 2–5 (for consistency with GPM-IMERG data).

### GPM-IMERG data
We use the daily Integrated Multi-satellitE Retrievals for GPM (IMERG) satellite retrievals of rainfall[66] for years 2001–2022. The IMERG data are re-gridded from their original 0.1° × 0.1° spatial resolution onto the ERA5 grid. We used daily mean precipitation data.

### Calculation of wet-bulb temperature
Daily mean Twb is calculated using daily mean dry-bulb temperature, dew-point temperature, and surface pressure from ERA5. We used the method of ref. 67 with Python code made available by ref. 68.

### Identification of heatwaves
Humid heatwaves are identified daily for land areas between 35°S and 35°N during the years 1993-2022:
- Hot humid days are identified separately in each grid cell. These are days when the daily mean Twb exceeds the 95th percentile for the local grid cell and exceeds a minimum threshold of 24 °C. The 95th percentiles are calculated separately for each grid cell using daily data during 1993–2022 and without using running means to avoid the pitfall highlighted by Brunner and Voigt[69]. The 95th percentiles capture local seasonality and regional variations in Twb.
- Hot humid days are aggregated into spatially and temporally contiguous heatwave events using a 3-dimensional (3D) connected components algorithm. The algorithm identifies clusters of contiguous grid cells and days that meet the criteria for hot, humid conditions. Twenty-six points of connectivity are recognised, which means that neighbouring grid cells are considered connected if they share an edge, face, or vertex in 3D (latitude, longitude, and time). The algorithm is available in Python from https://pypi.org/project/connected-components-3d/[70].
- The heatwave events are filtered by imposing a minimum duration of 3 days in each grid cell, and a minimum spatial extent of 3 grid cells (-1900 km²) throughout the lifetime of each heatwave.
- For consistency with the available GPM-IMERG data, heatwave data for years 2001–2022 is used in the analysis of the relationship between heatwaves and rainfall.
- We follow the practice of the World Meteorological Organization (WMO) and use a 30-year period (1993–2022) to calculate climatologies for heatwaves and other meteorological variables.

### Identification of rainy days
To compare heatwaves with rainfall we categorised each day in 2001–2022 into two categories using precipitation data from GPM-IMERG. The category with higher rainfall (High_Rain) includes days when rainfall exceeds 5 mm/day on that day, or the previous day, or on both days. The category with lower rainfall (Low_Rain) includes days when rainfall does not exceed 5 mm/day on both that day and the previous day. The 5 mm/day threshold produced the clearest results for the relative risk of heatwaves on High_Rain versus Low_Rain days. This threshold is physically meaningful; it is the median daily rainfall rate in GPM-IMERG (2001-2022) during the humid heatwave season in regions with humid heatwaves. The 5 mm/day threshold, therefore, represents above-average rainfall, which is sufficiently heavy to promote surface evaporation.

### The empirical probability of heatwave occurrence for enhanced and suppressed rainfall days
For each grid cell, we calculate the probability of heatwaves starting on High_Rain days:

$$\text{Pr}(\text{HW}|\text{High\_Rain}) = \frac{\text{N1}}{\text{N2}} \qquad (1)$$

where N1 is the number of High_Rain days on which a heatwave started, and N2 is the total number of High_Rain days. Similarly, we calculate the probability of heatwaves starting on Low_Rain days:

$$\text{Pr}(\text{HW}|\text{Low\_Rain}) = \frac{\text{N3}}{\text{N4}} \qquad (2)$$

where N3 is the number of Low_Rain days on which a heatwave started, and N4 is the total number of Low_Rain days.

The statistical significance of the difference between Pr(HW, |, High_Rain) and Pr(HW, |, Low_Rain) is tested following the method of Welty et al.[71]. We use a normal distribution approximation to the binomial test. The significance test is two-tailed and the significance level 5%.

The relative risk, sometimes referred to as the risk ratio, is calculated using one of two formulae:

$$\text{Relative Risk} = \frac{\text{Pr}(\text{HW}|\text{High\_Rain})}{\text{Pr}(\text{HW}|\text{Low\_Rain})} \qquad (3)$$

or

$$\text{Relative Risk} = \frac{\text{Pr}(\text{HW}|\text{Low\_rain})}{\text{Pr}(\text{HW}|\text{High\_Rain})} \qquad (4)$$

Equation (3) is used for grid cells where Pr(HW|High_Rain) is greater than Pr(HW|Low_Rain), and Eq. (4) is used where Pr(HW|Low_Rain) is greater than Pr(HW|High_Rain). We use Eqs. (3) and (4) for presentational purposes so that the relative risk varies over the same range of values (i.e., greater than 1) in all grid cells and all heatwave/rainfall regimes.

### Ecosystem Limitation Index (ELI)
The ELI[39] is the difference between two correlation coefficients:

$$\text{ELI} = \text{correlation}(\text{SLHF}', \text{SM}') - \text{correlation}(\text{SLHF}', \text{SSRD}') \qquad (5)$$

where SLHF′, SM′, and SSRD′ are daily anomalies from daily mean climatology for the surface latent heat flux, soil moisture, and surface SW radiation downwards. ELI is calculated daily using a 7-day moving window. We use daily anomalies and a 7-day window to capture variations in the surface environment on the time scale of humid

**Table 2 | Rules for the categorisation of grid cells to rainfall-ELI regimes**

| Regime | ELI | Relative risk | Probability of a heatwave given Low_Rain or High_Rain days |
|---|---|---|---|
| 0 | > 0 | Eq. 4 | Pr(HW\|Low_Rain) > Pr(HW\|High_Rain) |
| 1 | > 0 | Eq. 3 | Pr(HW\|High_Rain) > Pr(HW\|Low_Rain) |
| 2 | <0 | Eq. 4 | Pr(HW\|Low_Rain) > Pr(HW\|High_Rain) |
| 3 | <0 | Eq. 3 | Pr(HW\|High_Rain) > Pr(HW\|Low_Rain) |

heatwaves. We use a Spearman rank correlation to avoid assuming linear relationships.

The statistical significance of the heatwave composite ELI values was tested using the Wilcoxon signed rank test. This test was applied to the differences between corr(SLHF′, SM′) and corr(SLHF′, SSRD′) from values taken two days before each heatwave. Significance was determined at the 5% significance level.

### Categorisation of grid cells to rainfall-ELI regimes
The categorisation of grid cells to rainfall-ELI regimes is shown in Fig. 2c. This categorisation is based on the ELI values and relative risk (Table 2). Relative risk is calculated using the Pr(HW|High_Rain) and the Pr(HW|Low_Rain) (see The empirical probability of heatwave occurrence for enhanced and suppressed rainfall days above). For Regimes 1–3, both the ELI and the difference between Pr(HW|High_Rain) and Pr(HW|Low_Rain) are statistically significant at the 5% level. Grid cells categorised as not significant in Fig. 2c fail to meet the 5% significance level for either the ELI or the difference between Pr(HW|High_Rain) and Pr(HW|Low_Rain), or both.

### Wet-bulb temperature limitation index (TwbLI)
The Twb limitation index (TwbLI) is motivated by the ELI[39]. This index is analogous to the ELI but replaces surface latent heat flux anomalies with Twb anomalies in its calculation. TwbLI captures the competing influences of soil moisture and surface SW radiation downwards on Twb by comparing the strength of their respective relationships with Twb anomalies.

The evapotranspiration term used by ref. 39. is replaced by Twb. TwbLI is the difference between two correlation coefficients:

$$\text{TwbLI} = \text{correlation}\left(\text{Twb}', \text{SM}'\right) - \text{correlation}\left(\text{Twb}', \text{SSRD}'\right) \quad (6)$$

where Twb′, SM′, and SSRD′ are anomalies from daily mean climatology for Twb, soil moisture, and surface SW radiation downwards. TwbLI is calculated daily using a 7-day moving window. We use a Spearman rank correlation.

The statistical significance of the heatwave composite TwbLI values was tested using the Wilcoxon signed rank test applied in the same way as for the ELI.

### Calculation of the Heat Index
The Heat Index is calculated using the Rothfusz multiple regression formula published by the NOAA/National Weather Service[42]. In this formulation, the Rothfusz formula is modified for conditions of low or high relative humidity and when values of the Heat Index are below 80 °F (26.7 °C)[72].

### Data availability
The ERA5 reanalysis data are available from the Copernicus Climate Change Service https://doi.org/10.24381/cds.adbb2d47 (single level data)[73]. GPM-IMERG data are available from https://catalog.data.gov/dataset/gpm-imerg-final-precipitation-l3-1-day-0-1-degree-x-0-1-degree-v07-gpm-3imergdf-at-ges-dis[66]. HadISD sub-daily station data[47] are available from https://www.metoffice.gov.uk/hadobs/hadisd/v340_2023f/index.html.

### Code availability
Python 3 is used for data processing, analysis, and plotting. All custom codes use published implementations of standard methods and statistical techniques.

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

## Acknowledgements

Funding was provided by NERC grant Humid heat extremes in the Global (sub)Tropics (H2X): NE/X013618/1 (C.E.B., L.S.J., and J.H.M.) and NE/X013596/1 (G.C. and C.M.T). We acknowledge the Copernicus Climate Change Service[73] for making available the ERA5 data[64,65] and the NASA Goddard Earth Sciences Data and Information Services Center for the GPM-IMERG data[66]. We gratefully acknowledge Colin Raymond and Rob Warren for making available Python code for calculating Twb[68] and William Silversmith for Python code for the connected components

algorithm[70]. This work used JASMIN, the UK's collaborative data analysis environment (https://www.jasmin.ac.uk)[74].

## Author contributions

C.E.B., J.H.M. and C.M.T. designed the study. L.S.J. performed the analysis and wrote the first draft of the manuscript. C. E.B., G.C., J.H.M. and C.M.T. contributed to the interpretation of results and to writing of the final version of the manuscript.

## Competing interests

The authors declare no competing interests.
