## [Transparent Peer Review file · Nature Communications]

Daily rainfall variability controls humid heatwaves in the global tropics and subtropics

Corresponding Author: Dr Lawrence Jackson

Version 0:

Reviewer comments:

Reviewer #1

(Remarks to the Author)

Please see review comments attached.

Reviewer #2

(Remarks to the Author)

This is the review for the article "Humid heatwaves are controlled by daily rainfall variability". The article focuses on evaluating rainfall as a controlling mechanism for humid heatwaves for different regions. The authors identify differences depending on whether the region is energy-limited or moisture-limited. This study presents interesting findings. However, I wonder how these insights could be applied more broadly.

The impact of this study could be enhanced if the authors conduct additional analyses using climate models to evaluate the skill of GCMs in simulating this mechanism, assess projected changes in future periods, and explore the implications of these shifts. This is my major comment. I would like to see at least some, if not all, of these suggestions incorporated into the manuscript before I can recommend it.

A few other studies that author may consider citing.

- Fischer, E. M., & Knutti, R. (2013). Robust projections of combined humidity and temperature extremes. *Nature Climate Change*, 3(2), 126–130. <https://doi.org/10.1038/Nclimate1682>
- Rastogi, D., Lehner, F., & Ashfaq, M. (2020). Revisiting recent US heat waves in a warmer and more humid climate. *Geophysical Research Letters*, 47(9), e2019GL086736. <https://doi.org/10.1029/2019GL086736>
- Coffel, E. D., Thompson, T. R., & Horton, R. M. (2017). The impacts of rising temperatures on aircraft takeoff performance. *Climatic Change*, 144(2), 381–388. <https://doi.org/10.1007/s10584-017-2018-9>

Lines 169-171: How is the percentage of grids calculated? Does it represent the percentage of all land grid points globally?

The legend size and labels are hard to read in Figures especially Figure1d-e and Figure 2c.

404-406: Have authors considered using other metric such as apparent temperature instead of wet bulb temperature? What is the reason for using the wet bulb temperature?

411-413: "The 95th percentiles are calculated using daily data in each grid cell for all days during 1993-2022" Does this mean single 95th percentile value is calculated using 30*365 days for each grid cell? Why didn't the author consider calculating the 95th percentile for each year and then averaging across all years?

414-416: Additional clarification is needed here.

426-430: How did the authors determine the 5mm threshold? Shouldn't this threshold vary depending on the region?

477: It would be beneficial if the authors could make their code publicly available.

Reviewer #3

(Remarks to the Author)

This paper investigates the rainfall controls on daily humid heatwaves across the globe using gridded meteorological records by contrasting two regimes, the moisture-limited versus the energy-limited environment conditions. The work employs re-analysis based meteorological input, which itself has bias especially in arid region in capturing humidity term¹. Though sparse coverage in certain data constraint areas of the globe, a validation through observational records is required to claim robustness of obtained result.

My specific comments on the manuscript are below:

- 1) Line 51: Other than irrigation, proximity to the coast, see ref.2.
- 2) Line 57: "regional scale" could be elusive, for example, spatial heterogeneity could be very prominent within a continental or national scale.
- 3) Line 69 and elsewhere (for example, Fig. 1 caption line 126): The paper revolves back and forth between 90th percentile and 95th percentile of threshold selection in their write-up. The choice of threshold selection is not kept uniform throughout the paper. Also, they have used daily mean temperatures, whereas the daily maxima temperature should be used for heatwave analysis. This is because daily mean does not consider the abrupt rise for few hours, which can have fatal consequences for the exposed livelihoods
- 4) Line 94: Fig. 1A shows Himalayan region still experience very high frequency of humid heatwaves.
- 5) Line 97: regions with longest and persistent heatwaves are also experienced by desert areas of South Asia
- 6) Line 117: regional differences are expressed in a qualitative manner rather than quantitative ways.
- 7) Line 136: 5 mm/d rainfall threshold? Any references for heavy/moderately heavy rainfall?
- 8) Line 215: How these composites are constructed? Is it spatial average across all grid points?
- 9) Humid heat is often controlled by surface evaporation^{3,4}, however, the role of surface evaporation is not investigated among list of drivers in Fig. 4.
- 10) In Discussion: Lines 342-347: In Fig. 3SD: for south America and vast stretches of peninsular India, both with a strong tropical climate, no significant regime were identified, which is elusive. The same holds for northeast coast of the US, which has tropical climate.
- 11) Line 359: Choice of 24°C, as a fixed threshold could be erroneous without considering seasonality of humid heat across the globe. A daily variable threshold with a moving window approach should be considered to account for the seasonality. On page 20, Methodology, they have highlighted pitfalls of using running mean by citing Brunner & Voigt. But the reference never recommends using constant thresholds but rather they recommend for shortening window lengths.
- 12) Different Heat indices (HI) use different formulations and are not in agreement with each other. Further, the most widely use HI by Rothfus is based on polynomial approximation. In their paper, they have neither discussed which approach was followed for Heat index calculation nor the pitfalls of such an index.

In summary, I can't recommend that this manuscript to be accepted by NComm.

References

1. Raymond, C., Matthews, T. & Horton, R. M. The emergence of heat and humidity too severe for human tolerance. *Sci. Adv.* 6, eaaw1838 (2020).
2. Ganguli, P. & Merz, B. Observational Evidence Reveals Compound Humid Heat Stress-Extreme Rainfall Hotspots in India. *Earths Future* 12, e2023EF004074 (2024).
3. Bu, L., Zuo, Z., Zhang, K. & Yuan, J. Impact of Evaporation in Yangtze River Valley on Heat Stress in North China. (2023) doi:10.1175/JCLI-D-22-0573.1.
4. Zhang, Z. et al. Light rain exacerbates extreme humid heat. *Nat. Commun.* 15, 7326 (2024).

Version 1:

Reviewer comments:

Reviewer #1

(Remarks to the Author)

I thank the authors for their thorough responses to my comments, including their extended discussion of how these results fit into currently published literature.

My final suggestion for the paper relates to the use of the term "humidity" throughout the text. Some of the confusion in my initial reading was associated with assuming the authors were discussing relative vs. specific humidity, when they meant the opposite. I would suggest that the authors should change all mentions of general "humidity" to "relative humidity" or "specific humidity," where appropriate.

I believe that all of my other comments have been addressed.

Reviewer #3

(Remarks to the Author)

The revised paper has addressed and clarified several of my comments. However, there are a few minor issues, which can be addressed easily:

1) Regime 1 and Regime 2, show distinct differences in wet-bulb temperatures in HadISD observations relative to ERA5 (Figure RR2). While Regime 1 shows substantial differences within the time window $t \in (-1,5)$ -day, Regime 2 shows substantial differences across all time lags and leads, with ERA5 largely underestimate at-site measurements. This should be highlighted as caveats in the discussion section. Also, possible physical mechanisms behind this underestimation should be discussed.

2) The authors have not discussed the limitations of using Rothfusz's formulation in defining heat index, which is largely based on assumptions of polynomial approximation in its calculation, rather than physiological limits for withstanding heat stress, and its possible influence in their findings.

Response to Reviewers

Reviewer #1

In this manuscript, the authors evaluate the relationship between daily rainfall variability and humid heatwave dynamics. They particularly isolate the impacts of rainfall intensification and suppression based on categorizing grid cells by evapotranspiration regimes. The results of the study help advance our understanding of these relationships greatly, and are presented cleanly and concisely. Most of my comments are regarding result interpretation, communication, and contextualization with existing literature. I therefore recommend the authors complete minor revisions before the article be accepted.

We thank the reviewer for their thoughtful and thorough review. Our responses to your specific comments are set out below.

Title:

R1-1. o In the title, I would note that this manuscript focuses on just the tropics and subtropics.

We have changed the title to mention the tropics and subtropics.

Abstract:

R1-2. o Here as well, I would include mention that the manuscript will only focus on the tropics and subtropics.

We now include mention of the tropics and subtropics.

R1-3. o P2L9: Suggested change to “greater understanding of the meteorological drivers of extreme humid heat is urgently needed.”

We have inserted the text “ ... of extreme humid heat ...”.

R1-4. o P2L16: I think of adaptation as a modulator of vulnerability. Perhaps this sentence would make more sense if it mentioned vulnerability vs. exposure to extremes? I’m particularly thinking of the IPCC venn diagrams (see https://www.ipcc.ch/srocc/chapter/technical-summary/ts-0-introduction/ipccsrocc-ts_4/)

We have replaced the term “adaptation” with explicit reference to vulnerability and exposure.

☐ Introduction:

R1-5. o P4L70: 24°C actually seems quite low, and I’m not sure if you could say it’s a threshold relevant to human health impacts. I know you do a sensitivity test with 26°C, so I would either leave out the final clause of this sentence or mention that you chose this lower threshold for the main analysis in order to retain statistical significance of the results.

We agree there is limited evidence for the 24°C threshold being relevant to the effect of humid heat on human health and, where it is relevant, it will apply to vulnerable groups such as the elderly (Vanos et al., 2023). We have changed the sentence to state that the 24°C threshold is applied to:

- exclude conditions likely low risk for most people. We have also added references to Vecellio et al. (2023) and Vanos et al. (2023). We point the reviewer to Figure 4 in these papers.

- and, as suggested by the reviewer, to ensure there are sufficient data for statistically significant results.

Vanos, J., Guzman-Echavarria, G., Baldwin, J.W. *et al.* A physiological approach for assessing human survivability and liveability to heat in a changing climate. *Nat Commun* **14**, 7653 (2023). <https://doi.org/10.1038/s41467-023-43121-5>.

Vecellio, D. J., Kong, Q., Kenney, W., L. & Huber, M. Greatly enhanced risk to humans as a consequence of empirically determined lower moist heat stress tolerance. *PNAS* **120**, 42, e2305427120 <https://doi.org/10.1073/pnas.2305427120> (2023).

R1-6. o P4L77: I would include one or two sentences in this paragraph about the ongoing debate between physiology/epidemiology/economics/climate science regarding using Tw as a heat stress metric. Many epidemiologists and economists disagree that this represents humans experience of heat stress well. I think it is fine to use for climate science applications for understanding humid heat dynamics, but the authors should be upfront about the limitations of using this variable for human health applications (which they are in the discussion section, but I think it's worth mentioning before the results, as well). Example citations for this would include Baldwin et al. 2023 (<https://doi.org/10.1289/EHP11807>) and Vanos et al. 2020 (<https://doi.org/10.1038/s41467-020-19994-1>)

We thank the reviewer for recommending citations. We have added a few sentences to acknowledge the limitations of Twb in representing human heat stress. We have added the following specific points to the text:

- The 35°C threshold for Twb is an upper limit and may underestimate the health impacts in some contexts (Vanos et al., 2020).
- Twb does not separately identify the contributions of temperature and humidity, a distinction which can be important for health impacts (Ivanovich et al., 2024).
- Epidemiological studies fail to identify evidence for the impact of humidity on human health (Baldwin et al., 2023) although epidemiological evidence is likely to develop under increasingly extreme future temperatures driven by climate change.

Baldwin, J. W., Benmarhnia, T., Ebi, K. L., Jay, O., Lutsko, N. J. & Vanos, J. K. Humidity's Role in Heat-Related Health Outcomes: A Heated Debate. *Environmental Health Perspectives* 131(5), 055001 1-14 <https://doi.org/10.1289/EHP11807> (2023).

Ivanovich, C. C., Horton, R. M., Sobel, A. H. & Singh, D. Subseasonal variability of humid heat during the South Asian summer monsoon. *Geophysical Research Letters* 51, e2023GL107382 <https://doi.org/10.1029/2023GL107382> (2024).

Vanos, J. K., Baldwin, J. W., Jay, O. & Ebi, K. L. Simplicity lacks robustness when projecting heat-health outcomes in a changing climate. *Nature Communications* 11, 6079 <https://doi.org/10.1038/s41467-020-19994-1> (2020).

Results:

R1-7. o P6: I think the detailed discussion of climatologies of humid heatwaves and their seasonal cycle is a nice addition to the literature.

Thank you, this was our motivation for including Figure 1 and the accompanying text.

R1-8. o P6L116: Seems to me this conclusion should only be regions which have simultaneous peak rainfall and peak humid heat? I.e., the cream colored grid cells?

We agree with the reviewer and have changed the text to refer specifically to the cream coloured grid cells.

R1-9. o P9L168: I think it would be beneficial to provide a more quantitative explanation for how the Regimes are categorized, even if just included in the Methods section at the end of the manuscript. For example, is Regime 1 grid cells which have a positive relative risk and a positive ELI?

Yes, Regime 1 includes grid cells that have a positive ELI and positive relative risk. As requested, we have added a sub-section under Methods (“Categorisation of grid cells to rainfall-ELI regimes”) which explains in more quantitative terms how the grid cells are categorised.

R1-10. Additionally, I am having trouble determining how Regime 3 is identified. I think it would be helpful to have this explained in the text, and especially to describe it in the same paragraph above where Regimes 1 and 2 are introduced before the start of the next section (“Contributions of temperature and humidity to humid heatwaves”)

For clarity, we prefer to describe Regime 3 in its own paragraph, positioned immediately after the description of Regimes 1 and 2 and before the start of the next section (“Contributions of temperature and humidity to humid heatwaves”). The updated paragraph explains in more detail how Regime 3 is identified (i.e., grid cells with negative ELI and positive relative risk). Additionally, a more detailed explanation of the regime categorisation is now provided in the methods section. We believe these changes make clearer how Regime 3 is identified.

R1-11. o P11L209: I wonder if there is a way to show heatwave length? Especially as the authors begin to discuss what happens “when the heatwaves start to dissipate” (P12L226), it is unclear to me where in the time series these comments refer to. This is particularly important to clarify as it is my understanding that the heatwaves are defined as multiday events, but do not need to be the same duration across events.

Your understanding is correct: the heatwaves are multiday events lasting a minimum of 3 days with durations varying between events. The median duration is 4 days and, by day 7 in Figure 3, approximately 79% of the heatwaves have dissipated. Heatwave durations do not vary greatly between the regimes with median durations of 4, 4, and 4.5 days for Regimes 1, 2, and 3 respectively.

To avoid complicating Figures 3 and 4, we prefer not to add heatwave duration to these figures. We have changed the text, however, to explain more clearly how the composites are calculated and describe the dissipation of the heatwaves using the statistics quoted above. We also now state explicitly that the heatwaves start to dissipate on day 3. We have updated Figure 1b to show the median heatwave durations rather than mean durations. Median durations are shorter and indicate more clearly the dissipation of heatwave events over the timescales shown in Figures 3 and 4.

R1-12. o P12L228: I find the characterization of Regime 3 as “decreased” rainfall at the start of the humid heatwave to be deceptive. It appears that it is really elevated rainfall (positive anomaly) which decrease to 0 at the start of the heatwave and then rise back up (positive anomaly). Is there a better word to call this? Something like “neutral”?

We thank the reviewer for the suggestion and have changed the description from “decreased” to “neutral”.

R1-13. o P12L240: This mention of cloud cover is important for human experience of heat stress, which also depends on direct solar radiation. This relates to my comment below regarding the Discussion section about the need to include more information about the impacts of the heat stress metric you've selected.

While we acknowledge the importance of cloud cover for the human experience of heat stress, we have not made changes to this specific paragraph. As part of the Results section, this paragraph focuses on the dynamics of heatwaves, which is the primary scope of our study, rather than health impacts. However, we have addressed this point in the updated Discussion section, elaborating on the human experience of heat stress, particularly in relation to incoming solar radiation and wind speed. We also draw attention to the role of clouds in Figure 4h and the relevance for the human experience of heat stress.

R1-14. o P13L270: Interesting finding and interpretation about the evolution of these relationships, very nicely described.

Thank you.

R1-15. o Figures 3-4: It would be beneficial to include some measure of significance for these time evolution plots, or at least discussion of this in the text. Some of these anomalies are very small and I would assume indicate that these relationships are not statistically significant.

We have added a 95% confidence interval to the composite anomaly time series in Figures 3 and 4. This shows which variables have statistically significant composite anomalies and the days on which the significant changes occur.

All anomaly time series in Figure 3 are statistically significant at the 5% level. The anomalies are particularly small, however, for 2m temperature in Regime 3. Although previously noted in the text, we now emphasise this point by stating explicitly that the anomalies are less than 0.5°C (the maximum anomaly occurs on day 1 and is 0.41°C).

We have changed the text describing the results for Regime 2 in Figure 4 to make clear that the anomalies for TCWV, surface sensible heat flux, and boundary layer height are very small.

R1-16. o P15L286: Could you clarify what the "highest 10% of Twb" means? I'm a bit confused on when the analysis is using 95th percentiles, 90th percentiles, and then what this new requirement means in contrast.

We have expanded the description in the text to improve its clarity. To improve consistency throughout the manuscript and reduce the potential for confusion, we have also changed the definition to use the top 5% of heatwaves as measured by the maximum daily mean Twb in the heatwaves.

To help you assess the clarity of the revised description, a step-by-step description of the calculations is provided below:

1. For each grid cell, determine the periods classified as humid heatwaves.
2. Within each humid heatwave in a grid cell, find the highest daily mean Twb.
3. Collect the maximum daily mean Twb values for all humid heatwaves in a grid cell.
4. Calculate the 95th percentile of these maximum values for the grid cell.
5. Identify the heatwaves in which the maximum daily mean Twb exceeds the 95th percentile of the maximum Twb values in the grid cell.
6. Compute mean statistics for Figure 5 using this subset of extreme heatwaves.

R1-17. o P15L299: Could you comment on why this timeline is so far extended? It surprises me that the elevated temperature or moisture is coming >7 days before the event. What do you think is maintaining these changes in the week to follow?

The elevated temperatures or humidity seven days before extreme humid heatwave events are primarily due to inter-annual variability and preconditioning by prior heatwaves, likely driven by large-scale environmental conditions.

There is notable inter-annual variability in the number of humid heatwaves across the land grid cells in our analysis, with 39% of these cells having statistically significant variance in annual heatwave counts at the 5% significance level. This variability is driven by phenomena such as ENSO, which modulates near-surface temperature and humidity on inter-annual timescales. Unpublished research on our ERA5 dataset further reveals that humid heatwaves are significantly more frequent during El Niño years than La Niña years.

Extreme humid heatwaves tend to occur more frequently in years with a higher overall frequency of heatwaves. In 75% of the land grid cells analysed, years with at least one extreme humid heatwave have a greater number of heatwaves overall compared to years without extreme events. Additionally, in 34% of extreme humid heatwave events, a preceding heatwave event occurs seven days prior to the start of the extreme heatwave event. These preceding heatwave events likely contribute to elevated temperatures and humidity, which are maintained by persistent large-scale environmental conditions favourable to heatwave formation.

R1-18. o P15L307: Should also mention that the impact of a humid heat event of one magnitude also depends upon how it's made up – whether it's very high temperatures in the presence of some moisture vs. very high moisture in the presence of moderate temperatures. Extreme humid heat of a given threshold is more dangerous when we experience the former (high T, moderate q). Citations to include are Vecellio et al. 2021 (<https://doi.org/10.1152/jappphysiol.00738.2021>) and Ivanovich et al. 2024 (<https://doi.org/10.1175/JAS-D-23-0072.1>)

We have updated the manuscript to include mention of the dependence of humid heatwave impact on the individual contributions from temperature and humidity.

R1-19. o Figure 5: I find it interesting that in each subset of events (climatology, all heatwaves, most intense heatwaves), Regime 2 exhibits higher overall Twb than Regime 1. This seems counter to the findings in Ivanovich et al. 2024, for example, which suggest that most extreme humid heat events at the highest magnitudes are experienced due to elevated moisture. Could you comment on why you think there is a disagreement here?

You raise a very interesting issue, and we appreciate the opportunity to discuss the apparent differences between our findings and those of Ivanovich et al. (2024). Your comment highlights two key points: (1) why extreme humid heat events appear to differ in their dependence on elevated moisture, and (2) why Twb is greater in Regime 2 than in Regime 1.

1. Extreme Humid Heat Events and Elevated Moisture

The apparent differences between our results and those of Ivanovich et al. (2024) may be partly explained by the use of different measures of moisture. Ivanovich et al. (2024) use specific humidity to calculate their “stickiness” metric, whereas our results are presented using relative

humidity. Importantly, specific humidity is increased when temperature is increased under conditions of constant relative humidity. For example, in Figure 5 of our manuscript, Twb heatwaves in Regime 2 appear to be primarily temperature-driven, as there is only a small difference between climatological relative humidity and relative humidity during heatwaves. However, the large increase in temperature, coupled with near-constant relative humidity, implies that specific humidity is substantially above climatology during these events.

Ivanovich et al. (2024) present results for four regions, two of which (the Persian Gulf and South Asia) overlap geographically with our study. In the Persian Gulf region (likely corresponding to Regime 1 in our study), Ivanovich et al. (2024) find that extreme events are driven by increases in both specific and relative humidity, with little change in temperature compared to climatology. Similarly, in Figure 5 of our study, we find that extreme events in Regime 2 are associated with elevated relative humidity and only minor temperature changes, suggesting broad consistency between the two studies.

For South Asia (likely corresponding to Regime 2 in our study), Ivanovich et al. (2024) show that extreme events are driven by increases in both specific humidity and temperature. Relative humidity, however, differs little between extreme and all day events with the change approximately parallel to the relative humidity isopleths. This behaviour aligns with our findings in Figure 5, where relative humidity differs little between climatology and extreme events, but temperature is significantly elevated during extreme events.

2. Twb in Regime 2 vs. Regime 1

The observation that Twb is greater in Regime 2 than in Regime 1 is apparently inconsistent with the results presented in Figure 4 of Ivanovich et al. (2024). They report higher Twb for extreme events in the Persian Gulf region compared to South Asia. In contrast, our results in Figure 5 show extreme Twb to be greater for Regime 2 (which includes South Asia) than for Regime 1 (which includes parts of the Persian Gulf). The comparison of Regime 1 with the Persian Gulf and Regime 2 with South Asia should be interpreted with caution. Regime 1 includes parts of West Africa and North West India (amongst others) and 2 includes much of South America, tropical Africa, and northeast China. Further, the difference between Twb for extreme events in Regimes 1 and 2 is small (Figure 5).

Differences in Data and Methods

While the distinctions between specific and relative humidity partly explain the apparent differences between the two studies, several methodological differences may also be crucial.

- **Meteorological Data Sources:** Ivanovich et al. (2024) use station observations, whereas we use reanalysis data. In our validation of ERA5 reanalysis data against weather station observations, however, we found good overall consistency between ERA5 reanalysis and HadISD station observations for wet-bulb and 2m temperatures.
- **Definitions of Twb:** Ivanovich et al. (2024) analyse daily maximum Twb, whereas we focus on daily mean Twb. This distinction not only affects the magnitude of Twb but also likely alters the relative contributions of temperature and humidity to Twb.
- **Time Periods:** Differences in the time periods analysed further contribute to the apparent differences in findings.

Revisions to the Manuscript

In light of these thought-provoking issues, we have reviewed the language used in our manuscript regarding the contributions of temperature and humidity to humid heatwave events. We do not refer to temperature-driven or humidity-driven events. We have made one change in the section “The most extreme humid heatwaves”, where we have updated the description of Regime 2 extreme heatwaves to note that specific humidity, as well as temperature, is elevated.

Discussion:

R1-20. o P17L338: Could you provide a bit more context in the main text about your calculation of TwbLI? I’m having a bit of trouble with the interpretation in this paragraph even after reading the section in the Methods about it.

ELI captures the competing influences of soil moisture and surface SW radiation downwards on surface latent heat fluxes. To test our findings that surface energy- and moisture-limitations modulate the relationship between rainfall and humid heat, we defined a wet-bulb temperature energy-moisture limitation index (TwbLI). TwbLI captures more directly than ELI the competing influences of soil moisture and surface SW radiation downwards on Twb.

We have amended the relevant paragraph in the discussion section and the description in the methods section to include more explanation of the interpretation of the TwbLI.

R1-21. o P18L365: I think this paragraph needs to acknowledge that we know that humans’ experience of heat stress also depends on incoming solar radiation and wind speed. This is best represented by variables such as wet bulb globe temperature WBGT or the Universal Thermal Climate Index (UTCI). While I think that Twb is a fine choice for the analysis presented here, the relationship between precipitation and humid heat is definitely modulated by cloudiness (as you show in Figure 4) and wind speed. This inherently cannot be addressed by looking at Heat Index or 2m dry bulb temperature, which also do not incorporate these additional variables relevant to heat stress. A few additional sentences here to discuss this as a limitation in terms of how these results can be interpreted from a human health standpoint are important. Potential citations to help could include Grundstein and Vanos 2021(<https://doi.org/10.1136/bjsports-2020-102920>) and Budd 2008 (<https://doi.org/10.1016/j.jsams.2007.07.003>)

We have updated this paragraph to address the limitations of our research regarding human health and heat stress. We specifically discuss the influence of incoming solar radiation and wind speed, as highlighted in your comment, and have incorporated the references you kindly suggested (Budd 2008; Grundstein and Vanos, 2021).

Budd, G. M. Wet-bulb globe temperature (WBGT)—its history and its limitations. *Journal of Science and Medicine in Sport* **11**, 1, 20 – 32 <https://doi.org/10.1016/j.jsams.2007.07.003> (2008).

Grundstein, A. & Vanos, J. There is no ‘Swiss Army Knife’ of thermal indices: the importance of considering ‘why?’ and ‘for whom?’ when modelling heat stress in sport. *British Journal of Sports Medicine* **55**, 15, 822-824 <https://doi.org/10.1136/bjsports-2020-102920> (2021)

Reviewer #2 (Remarks to the Author):

This is the review for the article “Humid heatwaves are controlled by daily rainfall variability”. The article focuses on evaluating rainfall as a controlling mechanism for humid heatwaves for different regions. The authors identify differences depending on whether the region is energy-limited or moisture-limited. This study presents interesting findings. However, I wonder how these insights could be applied more broadly.

R2-45. The impact of this study could be enhanced if the authors conduct additional analyses using climate models to evaluate the skill of GCMs in simulating this mechanism, assess projected changes in future periods, and explore the implications of these shifts. This is my major comment. I would like to see at least some, if not all, of these suggestions incorporated into the manuscript before I can recommend it.

We thank the reviewer for their thoughtful review. We agree that the proposed analyses are important and would greatly enhance understanding of the mechanisms and their implications in a changing climate. However, our current study is focused on analysing the observed relationships and dynamics in the recent climate using reanalysis data which, to our knowledge, is lacking in the literature at the scale of the global (sub) tropics. We view this as a foundational step in understanding the humid heat-rainfall relationships. While incorporating climate model simulations and future projections is beyond the scope of the present study, we have plans for such work in the next stages of our project.

To acknowledge the importance of the reviewer’s suggestion, we have updated the final paragraph of the Discussion section to emphasise the opportunity for extending this work to climate change projections. We hope this provides clarity on the scope of our current work and its potential to inform further research.

Our responses to the detailed comments are set out below.

R2-22. A few other studies that author may consider citing.

Thank you for suggesting further studies to cite. We consider each study in turn below.

- Fischer, E. M., & Knutti, R. (2013). Robust projections of combined humidity and temperature extremes. *Nature Climate Change*, 3(2), 126–130. <https://doi.org/10.1038/Nclimate1682>

Although we appreciate the significance of this study, we have not added a citation because its focus is not directly relevant to the objectives and methods of our research. Fischer & Knutti (2013) focus on CMIP climate model outputs, which are not used in our study, and principally address climate change, which is beyond the scope of our study. Additionally, Fischer & Knutti (2013) use equivalent temperature and simplified wet-bulb globe temperatures to quantify the combined effects of temperature and humidity whereas we use wet-bulb temperature in our analysis.

- Rastogi, D., Lehner, F., & Ashfaq, M. (2020). Revisiting recent US heat waves in a warmer and more humid climate. *Geophysical Research Letters*, 47(9), e2019GL086736. <https://doi.org/10.1029/2019GL086736>

We have not added a citation for this study because its focus is the United States of America, whereas our study examines the global tropics and sub-tropics. Additionally, Rastogi et al. (2020) address climate change, which lies beyond the scope of our study.

- Coffel, E. D., Thompson, T. R., & Horton, R. M. (2017). The impacts of rising temperatures on aircraft takeoff performance. *Climatic Change*, 144(2), 381–388. <https://doi.org/10.1007/s10584-017-2018-9>

We have not added a citation for this study because its focus is on aircraft performance changes under climate change, neither of these issues are addressed in our study.

R2-23. Lines 169-171: How is the percentage of grids calculated? Does it represent the percentage of all land grid points globally?

The percentage of grid cells relates to land grid cells between latitudes 35°N and 35°S where humid heatwaves occurred during 1993-2022. We have amended the text to make this clear. We have also made the same change to the caption for Figure 2.

R2-24. The legend size and labels are hard to read in Figures especially Figure1d-e and Figure 2c.

We have increased the font size for legends and axis labels in all figures with specific attention paid to Figure 1 and Figure 2c. While the legends and axis labels are clearer in the individual figure files, they do not always appear so in the figures embedded in the manuscript document. We believe this is an issue with displaying figure files in Word documents and will not affect the figure quality in the published manuscript.

R2-25. 404-406: Have authors considered using other metric such as apparent temperature instead of wet bulb temperature? What is the reason for using the wet bulb temperature?

At the outset of our study, we considered available options for humid heat metrics and settled on wet-bulb temperature as our primary measure of humid heat for several reasons:

- It combines both temperature and humidity.
- It has similar thermodynamic properties to equivalent potential temperature (θ_e), making it robust for analysis on regional scales and for a study which focuses on the physical processes by which rainfall regulates the occurrence of humid heatwaves.
- It is relevant to human heat stress. Wet-bulb temperature provides an upper temperature limit for healthy, well-acclimatised human beings who have access to drinking water, shade, and a strong breeze.

We recognise there are alternative humid heat metrics that will exhibit different sensitivities to changes in temperature and humidity than wet-bulb temperature. To address this, we show the sensitivity of our results to different humid heat metrics in Figure S7 of the revised manuscript. Panel c shows the relative risk of heatwaves given rainfall based on the Heat Index (one apparent temperature metric) while panel d shows the same results based on 2m dry-bulb temperature.

R2-26. 411-413: “The 95th percentiles are calculated using daily data in each grid cell for all days during 1993-2022” Does this mean single 95th percentile value is calculated using 30*365 days for each grid cell? Why didn’t the author consider calculating the 95th percentile for each year and then averaging across all years?

Yes, the 95th percentiles used in our study are calculated using daily data for all days in all years (i.e., 30×365 days).

We have re-calculated the 95th percentiles using the reviewer’s suggested method. A comparison of percentiles from the two approaches is shown below in Figure RR1. For land grid cells where humid heatwaves occurred during 1993-2022, the method used in the manuscript yields percentile values a

little greater than the alternative method with a median difference of 0.09°C (Figure 1a, b). Figure 1c shows that the differences are broadly evenly distributed across the tropical land areas.

Based on this comparison, we have decided to retain our method for calculating the 95th percentiles. The results of our study would not change materially if we changed the method for calculating the 95th percentiles.

Figure RR1. Comparison of 95th percentiles for daily mean wet-bulb temperature. The percentiles used in the manuscript are calculated using the 95th percentile of values in all days and years and are referred to as “percentiles calculated over all years”. Alternatively, percentiles calculated separately for each of the 30 years and then averaged are referred to as “percentiles calculated as the mean of annual percentiles”.

R2-27. 414-416: Additional clarification is needed here.

We have clarified the description of the 3-dimensional connected components algorithm. Specifically, we now explain how the algorithm identifies spatially and temporally contiguous heatwave events using 26 points of connectivity, and we provide additional details on the definition of connectivity in 3-dimensions.

R2-28. 426-430: How did the authors determine the 5mm threshold? Shouldn't this threshold vary depending on the region?

We selected a threshold of 5 mm/day for enhanced rainfall days because that threshold produced the clearest results for the relative risk of heatwaves occurring on “High_Rain” versus “Low_Rain” days. The 5 mm/day threshold is physically relevant: it is the median daily rainfall during 2001-2022 in GPM-IMERG in the humid heatwave season in regions with humid heatwaves (Figure 1). We interpret the threshold as representing above average rainfall which is sufficiently heavy to promote evaporation.

Our study focuses on the global tropics and subtropics, emphasising processes operating at these large scales. While region-specific thresholds might enhance results in certain areas, they would complicate direct comparisons between different regions. By using a single threshold of 5 mm/day, we maintain consistency and comparability across the global tropics and subtropics. Despite using a common threshold for all regions, we find our results to be robust and compelling at the scale of our analysis.

We have added more justification for the 5 m/day threshold in the methods section of the manuscript. We present the results of sensitivity tests using different rainfall thresholds in Figure S6.

R2-29. 477: It would be beneficial if the authors could make their code publicly available.

While we understand the value of sharing code, we have decided not to make our code publicly available for this study because there is no novel methodology in our code. The analysis relies on standard data processing techniques and established statistical methods, which are implemented using widely available Python packages. For the calculation of wet-bulb temperature, we used the established method of Davies-Jones and Python code published by Raymond. For the identification of heatwave events, we used a 3D-connected components algorithm published in Python by Silversmith. Replicating the analysis should be straightforward using the details provided in the manuscript.

Davies-Jones, R. An efficient and accurate method for computing the wet-bulb temperature along pseudoadiabats. *Monthly Weather Review* **136**, 7, 2764-2785 DOI: 10.1175/2007MWR2224.1 (2008).

Raymond, C. https://github.com/cr2630git/wetbulb_dj08_spedup, accessed 16 October 2023.

Silversmith, W. connected-components-3d-3.12.1, <https://pypi.org/project/connected-components-3d/>, downloaded 15 January 2024.

Reviewer #3 (Remarks to the Author):

This paper investigates the rainfall controls on daily humid heatwaves across the globe using gridded meteorological records by contrasting two regimes, the moisture-limited versus the energy-limited environment conditions.

R3-44. The work employs re-analysis based meteorological input, which itself has bias especially in arid region in capturing humidity term. Though sparse coverage in certain data constraint areas of the globe, a validation through observational records is required to claim robustness of obtained result.

We have validated our analysis using hourly station observations from the HadISD dataset. In a new figure, Figure S2 in supplementary information, we compare daily mean wet-bulb temperatures from station observations and ERA5 grid cells with which the stations are “paired”. We show separate results for the three heatwave-rainfall regimes identified in this study and for the remaining stations at which the local heatwave-rainfall relationship is not statistically significant at the 95% confidence level. There is a substantial degree of overlap of the confidence intervals for the station and ERA5 composite time series. Further, important features of the ERA5 results are also present in the station observations:

1. The marked increase in wet-bulb temperatures (on day -1 and day 0);
2. The greatest increase in wet-bulb temperatures occurs in Regime 1;
3. The highest absolute wet-bulb temperatures occurs in Regime 2; and
4. The subsequent decrease in wet-bulb temperatures (from day +3).

We conclude that the results of Figure S2 validate both the wet-bulb temperatures derived from ERA5 data and the heatwave-rainfall relationships characterised by Regimes 1-3.

We have added a paragraph to the Discussion section summarising the rationale for this validation test and our findings. We have added a section in Supplementary Information summarising the methods we used.

We have repeated the comparison of ERA5 and HadISD station observations for 2m temperatures to check that the results for Twb are not masking large opposing errors in 2m temperatures and humidity. Reassuringly, there is a substantial degree of overlap of the confidence intervals for the station and ERA5 composite time series (Figure RR2).

For rainfall in our analysis, we used GPM-IMERG, a satellite-based observational product. We also tested its consistency with ERA5 precipitation. The relative risk results using ERA5 precipitation (Figure S7a) align closely with those obtained using GPM-IMERG precipitation (Figure 2a).

Figure RR2. Heatwave composite time series for daily mean 2m temperatures. The time series run from 7 days before the start of each heatwave (day -7) to 7 days after the start of each heatwave (day 7). Day 0 is the heatwave start day. The number of stations with sufficient data in each rainfall-heatwave regime is shown in the top right of each panel.

My specific comments on the manuscript are below:

R3-30. 1) Line 51: Other than irrigation, proximity to the coast, see ref.2.

Thank you for highlighting the work of Ganguli and Merz (2024). We have included a sentence describing the role played by proximity to the coast in the relationship between humid heat and rainfall.

R3-31. 2) Line 57: “regional scale” could be elusive, for example, spatial heterogeneity could be very prominent within a continental or national scale.

We agree that spatial heterogeneity can be very pronounced at continental or national scales. In our study, we use the term 'regional scale' to refer to areas smaller than the tropical and subtropical domains. We argue that the processes underlying humid heatwaves at the regional scale may not necessarily generalise to the broader tropical/subtropical scales, which underscores the value of conducting a global study focused on the tropics and subtropics.

We have amended the sentence to make clear the distinction between regional scales and the global tropics and subtropics.

R3-32. 3) Line 69 and elsewhere (for example, Fig. 1 caption line 126): The paper revolves back and forth between 90th percentile and 95th percentile of threshold selection in their write-up. The choice of threshold selection is not kept uniform throughout the paper.

We have amended our calculations and the manuscript to use the 95th percentile throughout. In Figure 1c, we now use the 95th percentile instead of the 90th percentile to represent the intensity of humid heatwaves. We have added 95% confidence intervals to Figures 3 and 4. In Figure 5, we now use the 95th percentile instead of the 90th to identify the extreme heatwave events (see R1-16).

R3-33. Also, they have used daily mean temperatures, whereas the daily maxima temperature should be used for heatwave analysis. This is because daily mean does not consider the abrupt rise for few hours, which can have fatal consequences for the exposed livelihoods

We believe that the choice of using daily mean temperatures or daily maximum temperatures depends on the purpose of the study. We concluded that daily mean temperature rather than daily maximum temperature is the most appropriate for this study. It captures the effects of both daytime heating and night-time cooling which are important for a study of processes at daily timescales. In terms of exposure to heat stress, daily mean temperatures capture accumulative exposure over longer periods than daily maxima and, importantly, it includes night-time temperatures which can make a substantial contribution to heat stress. In studies of the relationship between temperature and human health, daily mean temperatures are often used because they can provide robust results that are more easily interpreted in a policy context (Yang et al., 2012).

The reviewer makes a valid point, however. Heatwave definitions are frequently based on daily maximum temperatures and the most acute impacts of heat stress are often related to daily maximum temperatures. We have, therefore, repeated our analysis using daily maximum wet-bulb temperatures. The methods used are identical to those we used for daily mean temperature except that the minimum threshold was changed from 24°C to 24.75°C to approximately maintain the same number of heatwave days. We show the principle results in Figure S8 (which is equivalent to Figure 2 in the main manuscript) and Figure RR3 (which is equivalent to Figure 3a). The use of daily maximum temperature does not change the key results or conclusions of our study although the proportion of grid cells with statistically significant results is lower than for results based on daily mean temperatures.

Figure RR3. Reproduction of Figure 3a in the main manuscript but using daily maximum instead of daily mean wet-bulb temperatures.

We have updated the Discussion section to explain our rationale for using daily mean Twb, and also comment on our supplementary results using daily maximum Twb.

Yang, J., Ou, CQ., Ding, Y. *et al.* Daily temperature and mortality: a study of distributed lag non-linear effect and effect modification in Guangzhou. *Environ Health* **11**, 63 (2012).

<https://doi.org/10.1186/1476-069X-11-63>.

R3-34. 4) Line 94: Fig. 1A shows Himalayan region still experience very high frequency of humid heatwaves.

We have amended the text to refer to “areas within the Himalayan region ...”.

R3-35. 5) Line 97: regions with longest and persistent heatwaves are also experienced by desert areas of South Asia

We have updated Figure 1b to show the median heatwave durations (instead of mean durations) and have amended the accompanying text accordingly. The median duration is more representative of regional heatwave durations. A few long heatwaves in some locations greatly inflates the mean duration (e.g. near the eastern coast of South America).

We have updated the text to describe more precisely regional differences in humid heatwave durations.

R3-36. 6) Line 117: regional differences are expressed in a qualitative manner rather than quantitative ways.

We have amended the text to state explicitly that the comparison is qualitative. We have not updated Figure 1f to present a more quantitative comparison because, as explained in the first paragraph of the section “Rainfall variability as a driver of humid heatwave occurrence”, we subsequently proceed to present a quantitative analysis on daily timescales.

R3-37. 7) Line 136: 5 mm/d rainfall threshold? Any references for heavy/moderately heavy rainfall?

We selected a threshold of 5 mm/day for enhanced rainfall days because that threshold produced the clearest results for the relative risk of heatwaves occurring on “High_Rain” versus “Low_Rain” days. The 5 mm/day threshold is physically relevant: it is the median daily rainfall during 2001-2022 in GPM-IMERG in the humid heatwave season in regions with humid heatwaves (Figure 1). We interpret the threshold as representing above average rainfall which is sufficiently heavy to promote surface evaporation.

In the manuscript, we have amended the discussion on rainfall thresholds to include this additional explanation and interpretation. We have also included additional explanation in the methods section.

R3-38. 8) Line 215: How these composites are constructed? Is it spatial average across all grid points?

Grid cell heatwave composites are constructed by averaging daily mean time series spanning 7 days before to 7 days after the start of each heatwave. For grid cells where the heatwave-rainfall relationship is statistically significant (Figure 2), the composites are grouped into the regimes shown

in Figure 2d. The composite for each regime is then calculated as the average of grid cell composites within that regime.

We have added text to the beginning of the results sub-section “Contributions of temperature and humidity to humid heatwaves” to provide a more detailed explanation for the reader.

Regime composites are calculated by averaging the values of grid cells within each regime, without applying area weighting to individual grid cells. Since our analysis focuses on the characteristics of each regime rather than on contiguous geographic regions within those regimes, area weighting was not used. We tested the effect of area weighting, however, and found its impact to be negligible (Figure RR4).

Figure RR4. Heatwave composite time series from 7 days before the start of each heatwave (day -7) to 7 days after the start of each heatwave (day 7). Day 0 is the heatwave start day. Solid lines show daily mean values as presented in Figure 3 of the manuscript (i.e., with no area weighting of grid cells). Crosses show the equivalent values when the grid cells in each regime are area weighted.

R3-39. 9) Humid heat is often controlled by surface evaporation^{3,4}, however, the role of surface evaporation is not investigated among list of drivers in Fig. 4.

While we do not explicitly use the term "surface evaporation," we investigate its role through surface latent heat fluxes, which are equivalent but expressed in terms of energy flux rather than moisture flux. We prefer to show the surface latent heat flux because it provides consistency with other terms in the surface energy budget investigated in Figure 4; namely the sensible heat flux and downwelling radiation fluxes.

R3-40. 10) In Discussion: Lines 342-347: In Fig. 3SD: for south America and vast stretches of peninsular India, both with a strong tropical climate, no significant regime were identified, which is elusive. The same holds for northeast coast of the US, which has tropical climate.

We acknowledge the reviewer's observation regarding the absence of statistically significant regimes in regions such as South America, peninsular India, and the southeast coast of the US, despite their tropical climate. There are several potential reasons for this outcome:

- **Insufficient Heatwave Events:** In some regions, the number of heatwave events may be too small for the underlying signal to emerge from the noise introduced by sub-seasonal and inter-annual variability. This can hinder statistical significance despite the presence of physical processes linking heatwaves and rainfall.
- **Regional Variability in Heatwave-Rainfall Relationships:** The specific heatwave-rainfall relationships identified in many regions may not prevail universally. It is plausible that, in some locations, other processes dominate the dynamics of humid heatwaves, leading to weaker or different associations with rainfall.
- **Seasonal Cycle Effects:** The relationship between heatwaves and rainfall may vary at different times in the seasonal cycle. For example, in parts of peninsular India, the local environment may be moisture-limited during the pre-monsoon period but energy-limited after the onset of the monsoon. This seasonal variation could obscure consistent heatwave-rainfall associations when analysed over the entire season. Additionally, in some regions, such as parts of India, central Africa, and South America, our analysis shows that the humid heatwave season spans several months. This extended season further increases the likelihood of intra-seasonal variations in heatwave dynamics, which may dilute the statistical coherence of a single relationship over the entire period.

Although the reviewer raised this point in relation to Figure S3 in the supplementary information, it applies equally to our main results in Figure 2. Consequently, we have updated the paragraph discussing our Figure 2 results in the Discussion section to address these issues in greater detail.

R3-41. 11) Line 359: Choice of 24°C, as a fixed threshold could be erroneous without considering seasonality of humid heat across the globe. A daily variable threshold with a moving window approach should be considered to account for the seasonality. On page 20, Methodology, they have highlighted pitfalls of using running mean by citing Brunner & Voigt. But the reference never recommends using constant thresholds but rather they recommend for shortening window lengths.

We appreciate the reviewer's comment, which highlights the need for clearer explanation of how humid heatwaves are defined in the main text. To summarise, we use the local 95th percentile of daily mean T_{wb} , with an absolute minimum of 24°C, to capture humid heat events that are both high for the locality and to exclude events that are unlikely to pose a threat to human health. The 95th percentile of daily mean T_{wb} is calculated locally for each grid cell, accounting for regional variations, and is based on all days of the year, inherently capturing the seasonal cycle.

While the methods section provides a detailed definition, we recognise that additional context in the introduction and discussion sections would help ensure the approach is clear to readers. To address this, we have revised the introduction to briefly summarise the definition of humid heatwaves and clarified the paragraph in the discussion section where the reviewer's comment highlighted this issue.

R3-42. 12) Different Heat indices (HI) use different formulations and are not in agreement with each other. Further, the most widely used HI by Rothfus is based on polynomial approximation. In their paper, they have neither discussed which approach was followed for Heat index calculation nor the pitfalls of such an index.

We have amended the methods section to outline the calculation of the Heat Index and provide a reference to the specific formulae. In the discussion section, we now discuss the limitations of Twb as a measure of human heat stress in greater detail.

In summary, I can't recommend that this manuscript to be accepted by NComm.

We appreciate the time and effort you have taken to review our manuscript. We hope that the revisions made to the manuscript, in response to comments from all three reviewers, have substantially improved its clarity, robustness, and overall contribution to the field.

References

1. Raymond, C., Matthews, T. & Horton, R. M. The emergence of heat and humidity too severe for human tolerance. *Sci. Adv.* 6, eaaw1838 (2020).

2. Ganguli, P. & Merz, B. Observational Evidence Reveals Compound Humid Heat Stress-Extreme Rainfall Hotspots in India. *Earths Future* 12, e2023EF004074 (2024).

R3-43. 3. Bu, L., Zuo, Z., Zhang, K. & Yuan, J. Impact of Evaporation in Yangtze River Valley on Heat Stress in North China. (2023) doi:10.1175/JCLI-D-22-0573.1.

Thank you for drawing this paper to our attention. We have included a reference to it in the discussion section.

4. Zhang, Z. et al. Light rain exacerbates extreme humid heat. *Nat. Commun.* 15, 7326 (2024)

R3-44. See the top of page 15.

Response to Reviewers

Reviewer #1 (Remarks to the Author):

I thank the authors for their thorough responses to my comments, including their extended discussion of how these results fit into currently published literature.

We thank the reviewer for considering our responses and providing additional comments. Our responses to your specific comments are set out below.

My final suggestion for the paper relates to the use of the term "humidity" throughout the text. Some of the confusion in my initial reading was associated with assuming the authors were discussing relative vs. specific humidity, when they meant the opposite. I would suggest that the authors should change all mentions of general "humidity" to "relative humidity" or "specific humidity," where appropriate.

We have revised the manuscript to address the reviewer's comment by specifying "specific humidity" when discussing Figure 3 and "relative humidity" when referring to Figure 5, as these terms more accurately reflect the variables depicted in these figures. We have left other uses of the term "humidity" unchanged, as we believe its general usage is more appropriate in those contexts. Additionally, in several instances, we adhere to the terminology used in the literature we cite, where "humidity" is used in a broader sense.

I believe that all of my other comments have been addressed.

Reviewer #3 (Remarks to the Author):

The revised paper has addressed and clarified several of my comments. However, there are a few minor issues, which can be addressed easily:

We thank the reviewer for considering our responses and providing additional comments. Our responses to your specific comments are set out below.

1) Regime 1 and Regime 2, show distinct differences in wet-bulb temperatures in HadISD observations relative to ERA5 (Figure RR2). While Regime 1 shows substantial differences within the time window $t \in (-1,5)$ -day, Regime 2 shows substantial differences across all time lags and leads, with ERA5 largely underestimate at-site measurements. This should be highlighted as caveats in the discussion section. Also, possible physical mechanisms behind this underestimation should be discussed.

To address the reviewer's comment, we have added a paragraph to the Discussion section and added the relevant figure to Supplement Information (previously it was only disclosed in the response to reviewers). The additional text is quoted below:

“A comparison of ERA5 reanalysis and the HadISD 2m temperature dataset shows broad agreement, although ERA5 exhibits a cold bias in both Regimes 1 and 2. This bias may result from factors such as the averaging of ERA5 grid cell temperatures versus station point measurements, elevation differences, unresolved local topography⁴⁸, and the influence of land use/land cover on station observations. Additionally, ERA5 has been shown to underestimate daily maximum temperatures and extremes in some regions^{49,50}. Unlike Regime 1, the cold bias in Regime 2 is consistent across all lead-lag times. Regime 2 primarily occurs in moist tropical environments and in one such environment, tropical oceans, ERA5 has been found to underestimate temperatures and overestimate humidity, possibly due to deficiencies in relative humidity-dependent entrainment⁵¹. This mechanism could potentially contribute to the consistent bias in Regime 2.”

2) The authors have not discussed the limitations of using Rothfusz's formulation in defining heat index, which is largely based on assumptions of polynomial approximation in its calculation, rather than physiological limits for withstanding heat stress, and its possible influence in their findings

We have revised the manuscript to address the reviewer's comment by including discussion of limitations in the Rothfusz formulation in the relevant paragraph of the Discussion section. The revised paragraph is quoted below:

“Although an assessment of human heat stress is beyond the scope of this study, we recalculated relative risk using the Rothfusz formulation of the Heat Index (Figure S8c) and also 2m dry-bulb temperature (Figure S8d) to contextualise our results using two widely used temperature metrics. In energy-limited regions where heatwaves are more likely to follow suppressed rainfall, we obtain similar but more strongly significant results using the Heat Index or 2m dry-bulb temperature. In contrast, in moisture-limited regions the strength of the relationship between heatwaves and rainfall is much weaker. This supports our choice of Twb for this study and highlights differences in the sensitivity of different heat stress measures to humidity^{58,25,20}. The Rothfusz formulation for the Heat Index is an empirical polynomial approximation based on Steadman's physiological data^{59,60} and is not a comprehensive model of human heat tolerance. Its validity is limited under extreme temperature and humidity conditions, particularly beyond its original dataset^{61,62}. Furthermore, both

the Rothfus Heat Index and 2m dry-bulb temperature overlook factors such as incoming solar radiation and variations in wind speed, which are critical in outdoor environments⁵⁵.”

In this manuscript, the authors evaluate the relationship between daily rainfall variability and humid heatwave dynamics. They particularly isolate the impacts of rainfall intensification and suppression based on categorizing grid cells by evapotranspiration regimes. The results of the study help advance our understanding of these relationships greatly, and are presented cleanly and concisely. Most of my comments are regarding result interpretation, communication, and contextualization with existing literature. I therefore recommend the authors complete minor revisions before the article be accepted.

- Title:
 - In the title, I would note that this manuscript focuses on just the tropics and subtropics.
- Abstract:
 - Here as well, I would include mention that the manuscript will only focus on the tropics and subtropics.
 - P2L9: Suggested change to “greater understanding of the meteorological drivers **of extreme humid heat** is urgently needed.”
 - P2L16: I think of adaptation as a modulator of vulnerability. Perhaps this sentence would make more sense if it mentioned vulnerability vs. exposure to extremes? I’m particularly thinking of the IPCC venn diagrams (see https://www.ipcc.ch/srocc/chapter/technical-summary/ts-0-introduction/ipcc-srocc-ts_4/)
- Introduction:
 - P4L70: 24°C actually seems quite low, and I’m not sure if you could say it’s a threshold relevant to human health impacts. I know you do a sensitivity test with 26°C, so I would either leave out the final clause of this sentence or mention that you chose this lower threshold for the main analysis in order to retain statistical significance of the results.
 - P4L77: I would include one or two sentences in this paragraph about the ongoing debate between physiology/epidemiology/economics/climate science regarding using T_w as a heat stress metric. Many epidemiologists and economists disagree that this represents humans experience of heat stress well. I think it is fine to use for climate science applications for understanding humid heat dynamics, but the authors should be upfront about the limitations of using this variable for human health applications (which they are in the discussion section, but I think it’s worth mentioning before the results, as well). Example citations for this would include Baldwin et al. 2023 (<https://doi.org/10.1289/EHP11807>) and Vanos et al. 2020 (<https://doi.org/10.1038/s41467-020-19994-1>)
- Results:
 - P6: I think the detailed discussion of climatologies of humid heatwaves and their seasonal cycle is a nice addition to the literature.
 - P6L116: Seems to me this conclusion should only be regions which have simultaneous peak rainfall and peak humid heat? I.e., the cream colored grid cells?

- P9L168: I think it would be beneficial to provide a more quantitative explanation for how the Regimes are categorized, even if just included in the Methods section at the end of the manuscript. For example, is Regime 1 grid cells which have a positive relative risk and a positive ELI?
 - Additionally, I am having trouble determining how Regime 3 is identified. I think it would be helpful to have this explained in the text, and especially to describe it in the same paragraph above where Regimes 1 and 2 are introduced before the start of the next section (“Contributions of temperature and humidity to humid heatwaves”)
- P11L209: I wonder if there is a way to show heatwave length? Especially as the authors begin to discuss what happens “when the heatwaves start to dissipate” (P12L226), it is unclear to me where in the time series these comments refer to. This is particularly important to clarify as it is my understanding that the heatwaves are defined as multiday events, but do not need to be the same duration across events.
- P12L228: I find the characterization of Regime 3 as “decreased” rainfall at the start of the humid heatwave to be deceptive. It appears that it is really elevated rainfall (positive anomaly) which decrease to 0 at the start of the heatwave and then rise back up (positive anomaly). Is there a better word to call this? Something like “neutral”?
- P12L240: This mention of cloud cover is important for human experience of heat stress, which also depends on direct solar radiation. This relates to my comment below regarding the Discussion section about the need to include more information about the impacts of the heat stress metric you’ve selected.
- P13L270: Interesting finding and interpretation about the evolution of these relationships, very nicely described.
- Figures 3-4: It would be beneficial to include some measure of significance for these time evolution plots, or at least discussion of this in the text. Some of these anomalies are very small and I would assume indicate that these relationships are not statistically significant.
- P15L286: Could you clarify what the “highest 10% of Twb” means? I’m a bit confused on when the analysis is using 95th percentiles, 90th percentiles, and then what this new requirement means in contrast.
- P15L299: Could you comment on why this timeline is so far extended? It surprises me that the elevated temperature or moisture is coming >7 days before the event. What do you think is maintaining these changes in the week to follow?
- P15L307: Should also mention that the impact of a humid heat event of one magnitude also depends upon how it’s made up – whether it’s very high temperatures in the presence of some moisture vs. very high moisture in the presence of moderate temperatures. Extreme humid heat of a given threshold is more dangerous when we experience the former (high T, moderate q). Citations to include are Vecellio et al. 2021 (<https://doi.org/10.1152/jappphysiol.00738.2021>) and Ivanovich et al. 2024 (<https://doi.org/10.1175/JAS-D-23-0072.1>)

- Figure 5: I find it interesting that in each subset of events (climatology, all heatwaves, most intense heatwaves), Regime 2 exhibits higher overall Twb than Regime 1. This seems counter to the findings in Ivanovich et al. 2024, for example, which suggest that most extreme humid heat events at the highest magnitudes are experienced due to elevated moisture. Could you comment on why you think there is a disagreement here?
- Discussion:
 - P17L338: Could you provide a bit more context in the main text about your calculation of TwbLI? I'm having a bit of trouble with the interpretation in this paragraph even after reading the section in the Methods about it.
 - P18L365: I think this paragraph needs to acknowledge that we know that humans' experience of heat stress also depends on incoming solar radiation and wind speed. This is best represented by variables such as wet bulb globe temperature WBGT or the Universal Thermal Climate Index (UTCI). While I think that Twb is a fine choice for the analysis presented here, the relationship between precipitation and humid heat is definitely modulated by cloudiness (as you show in Figure 4) and wind speed. This inherently cannot be addressed by looking at Heat Index or 2m dry bulb temperature, which also do not incorporate these additional variables relevant to heat stress. A few additional sentences here to discuss this as a limitation in terms of how these results can be interpreted from a human health standpoint are important. Potential citations to help could include Grundstein and Vanos 2021 (<https://doi.org/10.1136/bjsports-2020-102920>) and Budd 2008 (<https://doi.org/10.1016/j.jsams.2007.07.003>).